# Accumulation of storage proteins in plant seeds is mediated by amyloid formation

Kirill S. Antonets[1,2], Mikhail V. Belousov[1,2], Anna I. Sulatskaya[3], Maria E. Belousova[1], Anastasiia O. Kosolapova[1,2], Maksim I. Sulatsky[3], Elena A. Andreeva[2], Pavel A. Zykin[2], Yury V. Malovichko[1,2], Oksana Y. Shtark[1], Anna N. Lykholay[2], Kirill V. Volkov[2], Irina M. Kuznetsova[3], Konstantin K. Turoverov[3], Elena Y. Kochetkova[3], Alexander G. Bobylev[4], Konstantin S. Usachev[5], Oleg. N. Demidov[3,6], Igor A. Tikhonovich[1,2], Anton A. Nizhnikov[1,2]*

1 All-Russia Research Institute for Agricultural Microbiology (ARRIAM), St. Petersburg, Russia, 2 St. Petersburg State University, St. Petersburg, Russia, 3 Institute of Cytology, Russian Academy of Sciences, St. Petersburg, Russia, 4 Institute of Theoretical and Experimental Biophysics, Russian Academy of Sciences, Pushchino, Moscow Region, Russia, 5 Laboratory of Structural Biology, Institute of Fundamental Medicine and Biology, Kazan Federal University, Kazan, Russia, 6 INSERM UMR1231, UBFC, Dijon, France

* a.nizhnikov@arriam.ru

**Data Availability Statement:** All relevant data are within the paper and its Supporting Information files.

## Abstract

Amyloids are protein aggregates with a highly ordered spatial structure giving them unique physicochemical properties. Different amyloids not only participate in the development of numerous incurable diseases but control vital functions in archaea, bacteria and eukarya. Plants are a poorly studied systematic group in the field of amyloid biology. Amyloid properties have not yet been demonstrated for plant proteins under native conditions in vivo. Here we show that seeds of garden pea *Pisum sativum* L. contain amyloid-like aggregates of storage proteins, the most abundant one, 7S globulin Vicilin, forms bona fide amyloids in vivo and in vitro. Full-length Vicilin contains 2 evolutionary conserved β-barrel domains, Cupin-1.1 and Cupin-1.2, that self-assemble in vitro into amyloid fibrils with similar physicochemical properties. However, Cupin-1.2 fibrils unlike Cupin-1.1 can seed Vicilin fibrillation. In vivo, Vicilin forms amyloids in the cotyledon cells that bind amyloid-specific dyes and possess resistance to detergents and proteases. The Vicilin amyloid accumulation increases during seed maturation and wanes at germination. Amyloids of Vicilin resist digestion by gastrointestinal enzymes, persist in canned peas, and exhibit toxicity for yeast and mammalian cells. Our finding for the first time reveals involvement of amyloid formation in the accumulation of storage proteins in plant seeds.

## Introduction

Amyloids represent protein aggregates having an unusual structure formed by intermolecular beta-sheets and stabilized by numerous hydrogen bonds [1]. Such a structure called "cross-β" [2] gives amyloids the morphology of predominantly unbranched fibrils and unique physicochemical properties including (i) resistance to treatment with ionic detergents and proteinases,

**Funding:** This work (study of amyloid formation by Vicilin and its domains in vivo and in vitro) was supported by the Russian Science Foundation, grant 17-16-01100. Part of this work performed by A.I. Sulatskaya (analysis of polymorphism of protein aggregates under various conditions) was awarded by RF President Fellowship SP-841.2018.4. The funders had no role in study design, data collection and analysis, decision to publish, or preparation of the manuscript.

**Competing interests:** The authors have declared that no competing interests exist.

**Abbreviations:** Aβ, human amyloid-beta peptide; CD, circular dichroism; C-DAG, Curli-Dependent Amyloid Generator; CR, Congo Red; HFIP, 1,1,1,3,3,3-Hexafluoro-2-propanol; IVPD, in vitro protein digestibility assay; MS/MS, tandem mass spectrometry; Omp, outer membrane protein; PBS, phosphate buffered saline; PB, protein bodies; pI, isoelectric point; PSIA-LC-MALDI, proteomic screening and identification of amyloids—high-performance liquid chromatography—matrix-assisted laser desorption/ionization mass spectrometry; RT, room temperature; SDS, sodium dodecyl sulfate; SDS-PAGE, sodium dodecyl sulfate—polyacrylamide gel electrophoresis; SEM, scanning electron microscopy; TBS, tris-buffered saline; TEM, transmission electron microscopy; ThT, Thioflavin T; YFP, yellow fluorescent protein.

(ii) binding amyloid-specific dyes like Thioflavin T (ThT), and (iii) apple-green birefringence in polarized light upon binding with Congo Red (CR) dye [1,3].

The biological significance of amyloids is based on two aspects, namely, pathological and functional. Amyloid deposition is associated with the development of more than 40 incurable human and animal diseases including various types of amyloidoses and neurodegenerative disorders [4,5]. Nevertheless, amyloids may not be only pathogenic but functional as well [6]. A growing number of studies demonstrate that amyloids play vital roles in archaea [7], bacteria, and eukarya including humans [8]. Amyloids of prokaryotes fulfill mostly structural (biofilm and sheaths formation) and storage (toxin accumulation) functions [9]. In fungi, infectious amyloids called prions control heterokaryon incompatibility, multicellularity, and drug resistance [10–12]. In animals, amyloid formation is important for different functions including the long-term memory potentiation, melanin polymerization, hormone storage, and programmed necrosis [13]. Compared to other groups of organisms, plants remain to be poorly studied in the field of amyloid biology.

Notably, the "amyloid" term was initially introduced in 1838 by Matthias Schleiden to describe plant cell carbohydrates and attributed in 1854 by Rudolph Virchow to pathological protein deposits in human tissues [14]. Early studies performed in 1920s through the 1950s have led to hypotheses on the presence of the so-called "amyloids" in plant seeds [15]. However, these structures were found to be xyloglucans, the major cell wall matrix polysaccharides [16]. Nevertheless, recently, some plant proteins or their regions were shown to form fibrils with several properties of amyloids in vitro (in denaturing conditions [17,18], after the proteolytic digestion or other treatments—reviewed in the work by Jansens and colleagues and the work by Cao and Mezzenga [19,20]), suggesting that plants might form bona fide amyloids in vivo [21].

Previously, we performed a large-scale bioinformatic analysis of potentially amyloidogenic properties of plant proteins including all annotated proteomes of land plant species [22]. This screening demonstrated that seed storage proteins comprising the evolutionary conservative β-barrel domain Cupin-1 were rich in amyloidogenic regions in the majority of analyzed species [22]. Such proteins belonging mainly to 11S and 7S globulins [23] represent key amino acid sources for the growing seedlings, important components of human diet and major allergens [24]. We hypothesize that the amyloid formation could occur at seed maturation to stabilize storage proteins, thus preventing their degradation and misfolding during the seed dormancy. In order to test this hypothesis, we have analyzed whether amyloid proteins are present in seeds of an important agricultural crop and Mendel's genetic model, garden pea *Pisum sativum* L.

## Results

### The storage proteins form aggregates resistant to treatment with ionic detergents in pea seeds

Resistance to treatment with ionic detergents is a typical feature of amyloids, discriminating them from the majority of other nonamyloid protein complexes [25]. We decided to study whether storage proteins form detergent-resistant aggregates in plant seeds. For this purpose, we have used mature seeds of garden pea *P. sativum* L. genetic line *Sprint-2* collected at 30 days after the pollination when the accumulation of storage proteins is expected to reach its maximum [26]. We have extracted protein complexes resistant to treatment with ionic detergent sodium dodecyl sulfate (SDS, 1%) from pea seeds using a previously published method called Proteomic screening and identification of amyloids—high-performance liquid chromatography—matrix-assisted laser desorption/ionization mass spectrometry (PSIA-LC-MALDI)

[27] with several modifications (see "Materials and methods"). The obtained detergent-resistant protein fractions have been solubilized with formic acid and subjected to trypsinolysis followed by the reversed-phase high-performance liquid chromatography and mass spectrometry (see "Materials and methods"). As a result, we have identified all 3 major classes of seed storage proteins (vicilins, convicilins, and legumins) and several other proteins including heteropolymeric iron-binding Ferritin, biotin-containing protein SBP65, and drought stress response protein Dehydrin as detergent-resistant components of pea seeds (S1 Table). Among identified proteins, 7S storage globulins vicilins have demonstrated the highest mass-spectrometric scores, suggesting for their prevalence in detergent-resistant fraction of pea seeds (S1 Table). We have selected 47 kDa Vicilin for a detailed analysis of its amyloid properties in vivo and in vitro because its N- and C-terminal regions have been identified by tandem mass spectrometry (MS/MS) analysis indicating that full-length protein participates in the formation of detergent-resistant polymers (S1 Fig).

## Recombinant Vicilin and its domains, Cupin-1.1 and Cupin-1.2, form detergent-resistant fibrils in vitro that bind amyloid-specific dye ThT

The Vicilin protein contains 2 evolutionary conserved Cupin-1 domains called Cupin-1.1 (19–166 aa) and Cupin-1.2 (229–394 aa; Fig 1A) belonging to the Cupin β-barrel domain superfamily. These domains share very little at the level of amino acid sequence exhibiting only 21.8% identity and 33.0% similarity [28,29]. Moreover, they significantly differ in their physicochemical properties: isoelectric points (pI) of Cupin-1.1 and Cupin-1.2 are 5.92 and 4.32, correspondingly (pI of the full-length Vicilin is 5.18) [28]. Nevertheless, according to the predicted with I-TASSER server [30], both Cupin-1.1 and Cupin-1.2 as well as full-length Vicilin form β-sheet-rich barrels (Fig 1A). Interestingly, all potentially amyloidogenic regions predicted using AmylPred2 method [31] in the Vicilin protein were located within boundaries of the Cupin-1.1 (regions indicated as R1–R5) and Cupin-1.2 (R6–R9) domains (Fig 1A). Moreover, R6–R9 regions of Cupin-1.2 were tightly stacked in β-sheets (Fig 1A). Furthermore, AmyloidMutants program [32] has predicted the ability of Vicilin and both Cupins to form juxtaposed β-strands assembling into β-sheets (S1 Data).

To check the ability of Vicilin and its domains, Cupin-1.1 and Cupin-1.2, to aggregate in vitro, we produced the full-length (414 aa) C-terminally 6x-His tagged Vicilin (without the N-terminally processed signal peptide), Cupin-1.1, and Cupin-1.2, in the *E. coli* cells, extracted, and purified them. We have found that incubation of Vicilin, Cupin-1.1, and Cupin-1.2 proteins in 5 mM phosphate buffered saline (PBS) [pH 7.4]) for one day at room temperature (25˚C) with the constant stirring has caused the formation of typical amyloid fibrils only in the Cupin-1.2 sample. At the same time, Vicilin and Cupin-1.1 have formed mainly morphologically unstructured aggregates visualized by the transmission electron microscopy (TEM; Fig 1B, row 1). Prolonged incubation (2 weeks) has caused the formation of more structured Cupin-1.1 but not Vicilin aggregates (Fig 1B, row 2). The increase in temperature (50˚C) has led to the formation of more compact Vicilin aggregates (Fig 1B, row 3). The best results have been obtained by the usage of 1,1,1,3,3,3-Hexafluoro-2-propanol (HFIP) solvent at 37˚C for the proteins dissolution with its subsequent removal from the sample and incubation of dissolved proteins in the distilled water (Fig 1B, row 4). Similar conditions have been used previously in other works to obtain amyloid fibrils of human amyloid-beta peptide (Aβ) [33] and RopA and RopB proteins of the bacterium *Rhizobium leguminosarum* [34].

We demonstrated that Vicilin has only been partially presented by fibrils and contained major fraction of less structured aggregates at 7 days after HFIP evaporation. Meanwhile, almost all aggregates of Cupin-1.1 and Cupin-1.2 have exhibited fibrillar morphology typical

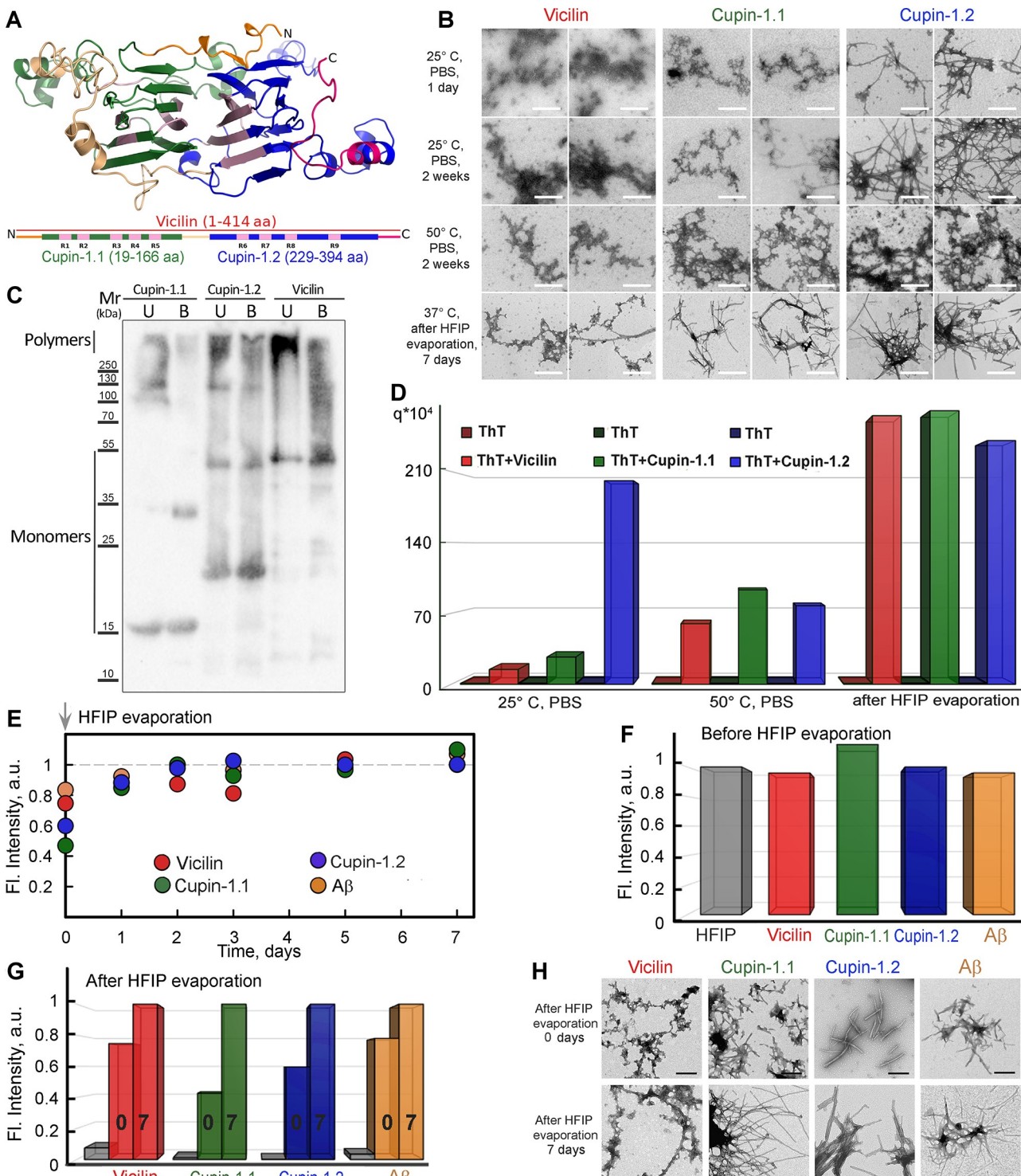

**Fig 1. Full-length Vicilin and its domains, Cupin-1.1 and Cupin-1.2, form detergent-resistant fibrils in vitro that bind ThT.** (A) The structure of the Vicilin monomer has been predicted by the I-TASSER server [30]. The domain structure of Vicilin is presented below with bars. The Cupin-1.1 domain is denoted by green, Cupin-1.2 with blue, and full-length Vicilin with red. The linkers are colored with ochre, light yellow, and magenta. N- and C-termini are shown. Predicted by AmylPred2 [31], amyloidogenic regions are indicated as R1–R9 and highlighted in pink. (B) TEM of the aggregates obtained in various conditions: in 5 mM PBS (row 1), with prolonged incubation (row 2) and temperature increase (row 3) as well as after dissolution in 50% HFIP followed by fibrillogenesis in water (row 4). The scale bars correspond to 500 nm. (C) The resistance of the Vicilin, Cupin-1.1, and Cupin-1.2 aggregates to treatment with cold (U, unboiled) and hot (B, boiled) 2% SDS. Molecular weights (kDa) are shown. (D) Fluorescence

quantum yield of amyloid-specific probe ThT in a free state in buffer solution (left bars) and bound to aggregates (right bars) have been determined by using the equilibrium microdialysis for the sample preparation. Conditions of aggregates preparation are the same as in Panel B. The colors that indicate the different proteins correspond to those in Panel B. (E) Normalized polymerization kinetics curves for Vicilin (red circles), Cupin-1.1 (green), Cupin-1.2 (blue), and Aβ (1–42 aa; orange) detected with the use of ThT. (F) Fluorescence of ThT in the free state in HFIP (gray color) and in the presence of monomeric proteins before HFIP evaporation (colors are similar to those in Panel E) normalized to free dye fluorescence. (G) ThT fluorescence in the water solution (gray color) and in the presence of aggregated proteins immediately (0) and 7 days (7) after HFIP evaporation normalized to the maximum value for each sample. Colors are similar to those in Panel F. (H) TEM images of Vicilin, Cupin-1.1, Cupin-1.2, and Aβ (1–42 aa) aggregates immediately and 7 days after HFIP evaporation. The scale bars correspond to 200 nm. Values for graphs (D–G) are listed in S4 Data. HFIP, 1,1,1,3,3,3-Hexafluoro-2-propanol; SDS, sodium dodecyl sulfate; TEM, transmission electron microscopy; ThT, Thioflavin T.

for amyloids (Fig 1B, row 4). We have investigated the resistance of the in vitro obtained Vicilin, Cupin-1.1, and Cupin-1.2 aggregates to treatment with ionic detergents. For this purpose, protein samples have been treated for 5 min with the sodium dodecyl sulfate—polyacrylamide gel electrophoresis (SDS-PAGE) sample buffer containing 2% SDS or have been boiled for 5 min with the same buffer containing 2% SDS and then have been loaded onto the gel and have been subject to SDS-PAGE followed by western blot analysis. The results of the experiment have demonstrated that all 3 proteins form aggregates resistant to the treatment with both cold and hot SDS with the part of aggregates dissolved after boiling (Fig 1C).

To confirm the assumption of the Vicilin, Cupin-1.1, and Cupin-1.2 fibril formation, we have studied the interaction of the obtained aggregates with ThT [35]. A unique feature of this dye is a very weak fluorescence in the free state in an aqueous solution and an intense fluorescence in the bound to fibrils state. Because the samples contained both ones bound to fibrils and free ThT, we have used an equilibrium microdialysis for preparing test solutions and the separation of photophysical characteristics of 2 different dye fractions [36]. The use of this approach and other special techniques, taking into account the contribution of the aggregates light scattering to the recorded absorption spectra [36] and the correction of the recorded fluorescence intensity to the primary inner filter effect [37], has enabled to calculate the fluorescence quantum yield of free ThT in the samples (the value of which turned out to be close to $10^{-4}$, which suits the literature data [38]) and the dye associated with the studied aggregates (Fig 1D).

We have found that ThT binds to all the tested samples, however, an increase in the fluorescence quantum yield of the bound dye in comparison to that for free ThT varies considerably in different samples (Fig 1D). The greatest increase in the fluorescence quantum yield is observed in the case of "typical" Cupin-1.2 fibrillar aggregates obtained in the phosphate buffer at 25˚C (more than 2 orders of magnitude). At the same time, ThT fluoresces significantly less when it is binding to Vicilin and Cupin-1.1 aggregates obtained in the same conditions (Fig 1D). It can be assumed that the main fraction of the bound dye in these samples interacts with aggregates (apparently with less compact and ordered compared to fibrillar ones) not specifically, which leads to an insignificant restriction of the intramolecular mobility of ThT fragments relative to one another, and, therefore, to a small increase in its fluorescence quantum yield. ThT bound to Vicilin, Cupin-1.1, and Cupin-1.2 aggregates obtained in the phosphate buffer at 50˚C has exhibited the similar intermediate means of the fluorescence quantum yields (Fig 1D) indicating their partially structured morphology (Fig 1B, row 3). Finally, ThT bound to the Vicilin, Cupin-1.1, and Cupin-1.2 fibrils obtained 7 days after HFIP evaporation at 37˚C has exhibited high fluorescence quantum yields (Fig 1D) confirming their highly structured spatial characteristics. Considering the obtained results, the latter conditions have been chosen for preparation of fibrils in vitro in further experiments.

Next, we have studied kinetics of amyloid fibrils formation under chosen conditions. After preliminary incubation of the proteins in 50% HFIP, the solvent has been removed from the sample by evaporation in a stream of nitrogen, and we have conducted the Vicilin, Cupin-1.1,

and Cupin-1.2 polymerization study using ThT (Fig 1E). To compare, similar experiments have been performed with one of the most characterized amyloids, Aβ (1–42 aa). We have found that fluorescence of ThT in the presence of monomeric proteins before HFIP evaporation has been almost the same as the free dye fluorescence (Fig 1F). This observation has proven the lack of the dye interaction with monomeric proteins. After the HFIP evaporation, a significant increase of ThT fluorescence has been detected (Fig 1G). Based on these results, we may conclude that fibrils of all tested proteins are formed immediately after the HFIP evaporation. This assumption has been proven by the TEM images of samples that have been obtained after HFIP evaporation (Fig 1H). Thus, the Vicilin plant protein and its Cupin-1.1 and Cupin-1.2 domains exhibit fibril formation kinetics in vitro that is very similar to those of Aβ (1–42 aa) in the chosen conditions.

## Fibrils of the full-length Vicilin, Cupin-1.1, and Cupin-1.2 possess amyloid properties

One of the important structural features of amyloids is their richness in β-sheets [2]. For determination of the secondary structure of the Vicilin, Cupin-1.1, Cupin-1.2, we recorded far-UV circular dichroism (CD) spectra of these proteins dissolved in 50% HFIP (Fig 2A, left) and in distilled water where these proteins were incubated for 7 days after HFIP evaporation to obtain fibrils (Fig 2A, right). A significant change in the CD spectra of the samples during fibrillogenesis has been detected. Samples of the Cupin-1.1, Cupin-1.2, and Vicilin fibrils obtained in distilled water 7 days after HFIP evaporation have demonstrated apparent negative shoulder around 220 to 230 nm [39] (Fig 2A, right) that allows us to suggest the essential content of β-sheets in their structure [40]. This spectral minimum has been less pronounced for Vicilin aggregates that does not contradict the results of TEM, indicating a lower content of highly ordered fibrillar structures in the Vicilin sample compared to the Cupin-1.1 and Cupin-1.2 samples (Fig 1).

For estimation of the protein secondary structure content, CDPro software containing 3 popular methods (CONTIN, SELCON3, and CDSSTR) [41] and several basic sets of proteins with a known secondary structure (that include from 37 to 56 soluble, membrane, and denatured proteins with different content of the secondary structure) have been used. The results obtained by different methods have been summarized and averaged in S2 Table. It was shown that the Cupin-1.1 and Cupin-1.2 proteins dissolved in 50% HFIP have exhibited relatively low β-sheet content (4%–12%) in comparison with Vicilin (39%; S2 Table). Further formation of the Vicilin, Cupin-1.1, and Cupin-1.2 fibrils in water 7 days after HFIP evaporation have resulted in the increase of the β-sheet content (40%–42%) in their structures that was approximately the same as for the Aβ (1–42 aa) fibrils (40%) used as the positive control (S2 Table).

Thereby, despite the differences in the recorded CD spectra, quantitative analysis has demonstrated high content of the ordered β-structure (β-sheets and β-turns) in the samples of the Cupin-1.1, Cupin-1.2, and Vicilin fibrils confirming their amyloid nature. Similar results have been obtained for amyloid fibrils formed from Aβ (1–42 aa). It can be stressed that the content of β-sheets in the bona fide amyloid fibrils formed by previously characterized amyloid proteins is often the same or even lower than it has been shown in our work for Vicilin, Cupin-1.1, and Cupin-1.2 [42–44].

Next, we have stained Vicilin, Cupin-1.1, and Cupin-1.2 aggregates with CR dye. When CR binds to amyloids it leads to the so-called "apple-green birefringence" in polarized light [45]. This effect is highly specific and for a long time has been considered the "gold standard" in the amyloid diagnostics [4]. The results of this test have demonstrated that Vicilin, Cupin-1.1, and Cupin-1.2 aggregates bind CR and exhibit birefringence (Fig 2B). Notably, despite the fact that

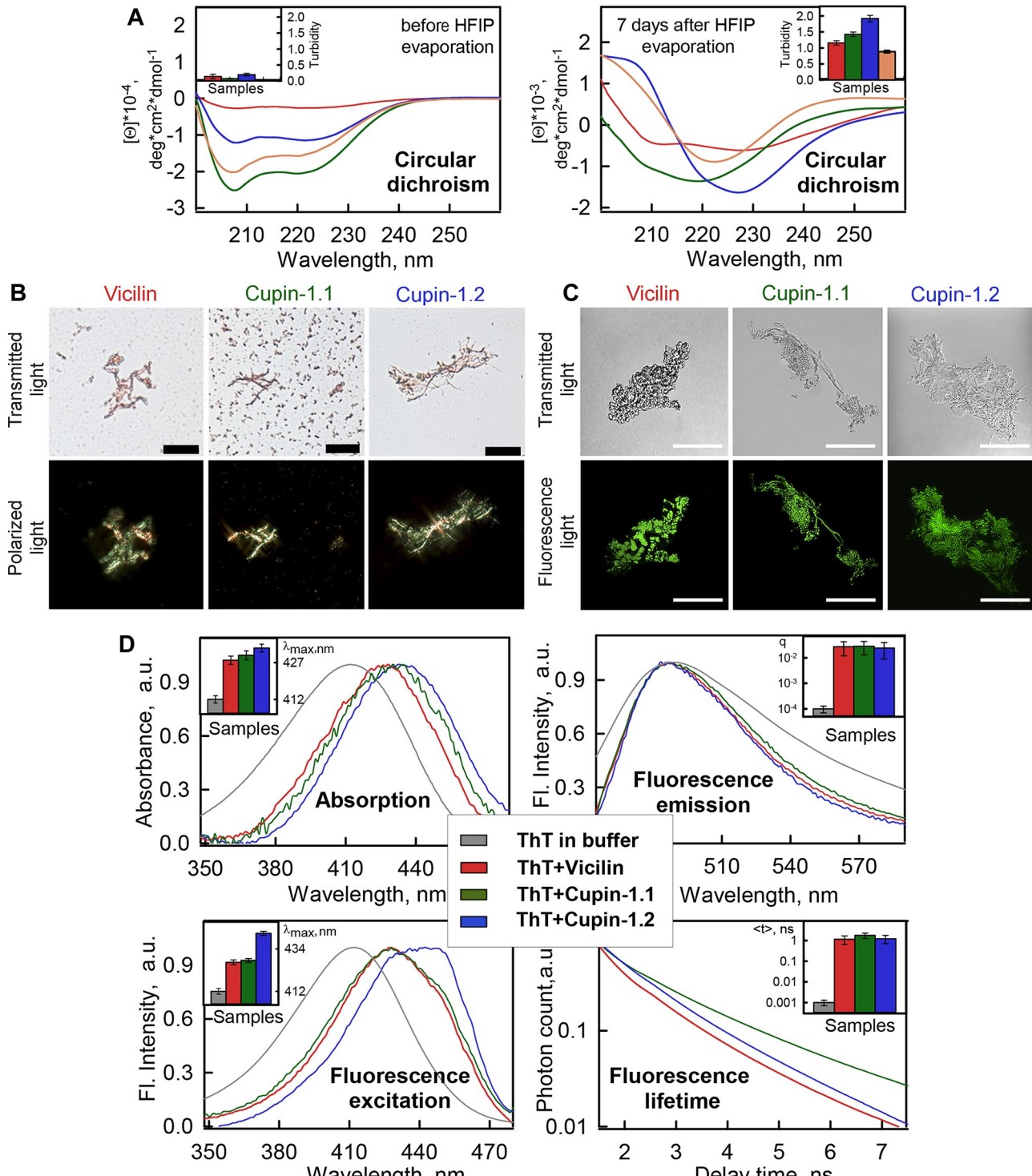

**Fig 2. Vicilin, Cupin-1.1, and Cupin-1.2 aggregates exhibit amyloid properties.** (A) CD analysis of the Vicilin (red), Cupin-1.1 (green), Cupin-1.2 (blue), and Aβ (orange) samples before (left panel) and after HFIP evaporation followed by incubation in water for 7 days (right panel). The inset shows the turbidity of the samples. (B) Polarization microscopy of the Vicilin, Cupin-1.1, and Cupin-1.2 aggregates stained with CR. The scale bar is equal to 20 μm. Top row: transmitted light; bottom: polarized light. (C) Confocal microscopy of the Vicilin, Cupin-1.1, and Cupin-1.2 aggregates stained with ThT. Fluorescence images of the ThT-stained aggregates (bottom row) and transmitted light images showing the presence of aggregates in the sample (top row) are presented. The scale bar is equal to 15 μm. (D) Absorption, fluorescence excitation, and emission spectra and fluorescence decay curves of the ThT bound to Vicilin, Cupin-1.1, and Cupin-1.2 aggregates. The insets of the corresponding panels show the position of the maxima of the

absorption and fluorescence excitation spectra, the values of the fluorescence quantum yield, and lifetime of the bound to fibrils dye. Decoding for the used colors is given in Panel D. Values for graphs (A) and (D) are listed in S4 Data. CD, circular dichroism; CR, Congo Red; HFIP, 1,1,1,3,3,3-Hexafluoro-2-propanol; ThT, Thioflavin T.

Vicilin aggregates obtained in the aforementioned experiments do not exhibit clear fibrillar morphology (Fig 1B), they contain a significant admixture of the amyloid fibrils exhibiting birefringence that is easily detected by the polarization microscopy (Fig 2B).

The amyloid nature of the studied samples has also been proven by their confocal microscopy in the presence of ThT. The dye has stained not only Cupin-1.1 and Cupin-1.2 fibrils but also less ordered Vicilin aggregates (Fig 2C). Using solutions prepared by the equilibrium microdialysis, we have determined the absorption, fluorescence and fluorescence excitation spectra, as well as the fluorescence decay curves of ThT bound with these aggregates (Fig 2D). We have shown the long-wavelength shift of the dye absorption and fluorescence excitation spectra when it binds to protein aggregates (Fig 2D, insets of left panels). The determined excitation spectra have a maximum at a wavelength of about 430 nm and longer-wavelength shoulder of about 450 nm, which indicates the existence of 2 different types of the dye-amyloid binding (Fig 2D, bottom left panel). A binding mode with a maximum of the absorption and fluorescence excitation spectra of about 450 nm was previously detected when ThT binds to amyloid fibrils formed from insulin and lysozyme [36,46]. Another binding type can be due to the fraction of the dye molecules associated with the less ordered and compact aggregates in the samples. The maximum of fluorescence spectra of ThT bound to Vicilin, Cupin-1.1, and Cupin-1.2 aggregates (Fig 2D, top right panel) coincides with the maximum of that of the free dye, as in the case of ThT interaction with amyloid fibrils formed from other amyloidogenic proteins [47]. In addition, we have shown a significant increase in the fluorescence quantum yield and lifetime (calculated using recorded fluorescence spectra and fluorescence decay curves, respectively) of the dye bound to tested aggregates (Fig 2D, right panels), which is also a characteristic feature of the ThT interaction with amyloid aggregates. Close values of the fluorescence quantum yields and lifetimes of ThT bound to Vicilin, Cupin-1.1, and Cupin-1.2 fibrils indicate a similarity of the dye binding sites, and therefore, similarity of structures of amyloid fibers formed by Vicilin and its domains, Cupin-1.1 and Cupin-1.2.

Thus, based on the wide array of experimental data we may conclude that the full-length Vicilin and both its domains Cupin-1.1 and Cupin-1.2 form amyloid aggregates in vitro. The fibril formation by Cupin-1.1 and Cupin-1.2 suggests their contribution to the amyloid formation by the full-length Vicilin. In contrast to both Cupin domains, Vicilin tends to form mixture of bona fide amyloid fibrils with less structured aggregates.

### Amyloid fibril formation of the full-length Vicilin can be efficiently seeded by the preincubated Vicilin or Cupin-1.2 fibrils

The "seeding" of amyloids is a process that represents triggering of the amyloid fibril formation by inoculation with preformed amyloid fibrils of the same or other proteins called "seeds." This process is mostly sequence-specific, and de novo forming fibrils exhibit close structural similarity or identity to the seeds [48,49].

We have aimed to investigate the ability of the Vicilin, Cupin-1.1, and Cupin-1.2 aggregates to induce the fibrillation of the full-length Vicilin. For this purpose, we have used seeds prepared from preincubated Vicilin, Cupin-1.1, or Cupin-1.2 aggregates (by protein dissolution in 50% HFIP followed by evaporation of HFIP and incubation in distilled water). It has turned out that Vicilin aggregates obtained by usage of the seeds on the basis of Cupin-1.1 aggregates do not differ in their morphology from those prepared in the absence of seeds (Fig 3A, middle

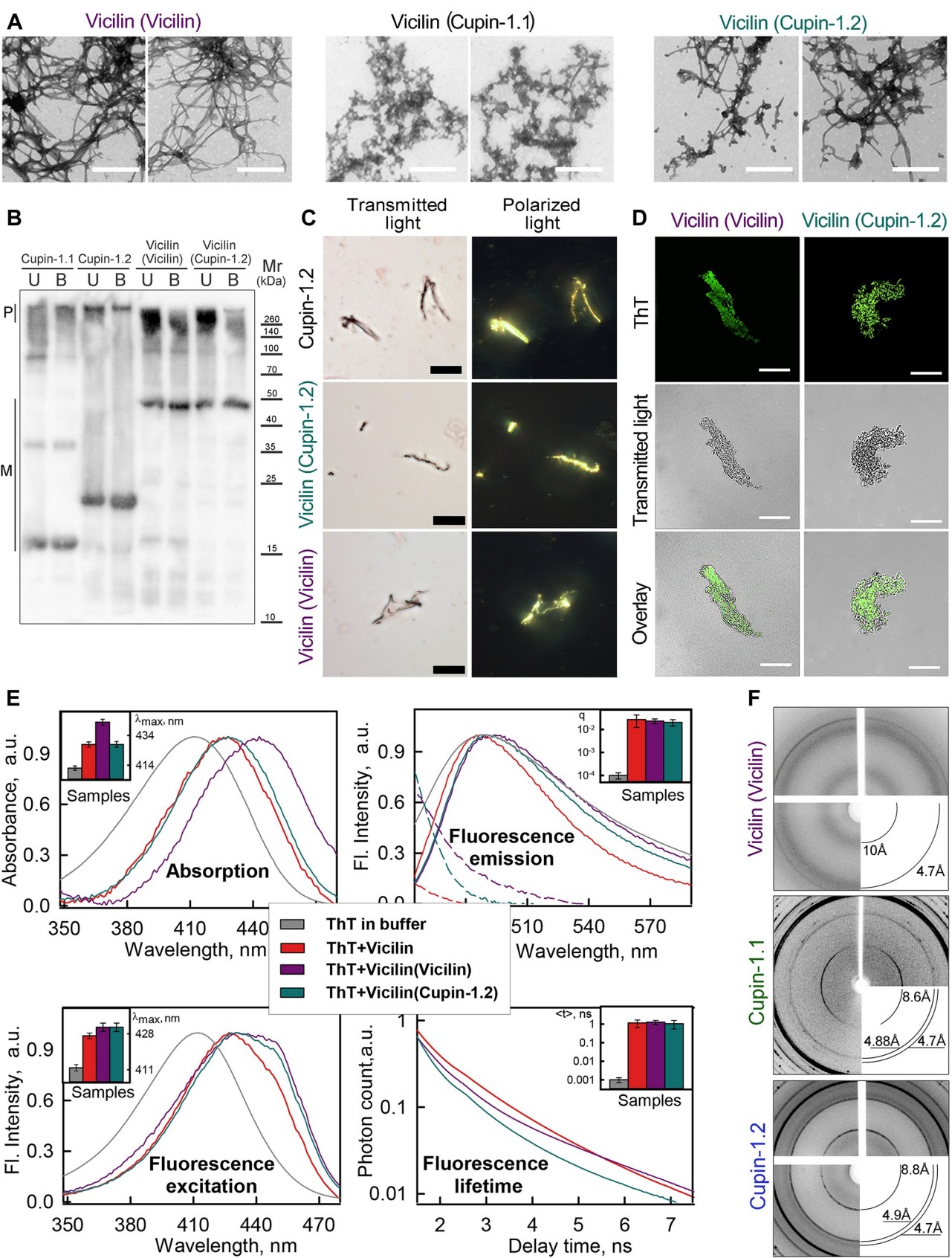

**Fig 3. Vicilin amyloid fibril formation is efficiently seeded by its aggregates or Cupin-1.2 fibrils.** Seeds used for each of the samples are indicated in parentheses. (A) TEM of the Vicilin fibrils obtained by seeding. The scale bar is equal to 500 nm. (B) Detergent resistance of the fibrils to cold (U, unboiled) and hot (B, boiled) 2% SDS. Respective molecular weights (kDa) are shown. Cupin-1.1 and Cupin-1.2 fibrils were used as control. (C) Birefringence of Vicilin fibrils stained with CR. Scale bar is equal to 20 μm. Left: transmitted light; right: polarized light. Cupin-1.2 fibrils were used as control. (D) Confocal microscopy of the fibrils stained with ThT. Fluorescence images of the ThT (top row), transmitted light images (middle row), and overlay (bottom row) are presented. The scale bar is equal to 15 μm. (E) Normalized to unity at the maximum absorption, fluorescence excitation and emission spectra and fluorescence decay curves of the ThT bound to Vicilin aggregates. The insets of the corresponding panels show the position of the maxima of the absorption and fluorescence excitation spectra, the values of the fluorescence quantum yield and lifetime of the bound to fibrils dye. By the dotted lines the long-wavelength regions of normalized absorption spectra (considering differences in the concentration of the bound dye) are presented. Decoding for the colors used is given in the inset of Panel E. (F) x-ray diffraction patterns of the lyophilized Vicilin, Cupin-1.1, and Cupin-1.2 fibrils formed in vitro. Shown are reflections in angstroms (Å). Values for graphs in Panel E are listed in S4 Data. CR, Congo Red; SDS, sodium dodecyl sulfate; TEM, transmission electron microscopy; ThT, Thioflavin T.

panel). At the same time, using the seeds from the Cupin-1.2 fibrils has resulted in more efficient formation of the Vicilin fibrils (Fig 3A, right panel). Finally, seeds from the Vicilin aggregates obtained in previous experiments have caused the formation of almost exclusively Vicilin fibrils with the visual absence of unstructured aggregates (Fig 3A, left panel).

We have studied the resistance of the obtained by seeding Vicilin fibrils to treatment with ionic detergents and have found that they are resistant to cold and hot SDS (Fig 3B). Staining of Vicilin fibrils with CR has demonstrated the clear apple-green birefringence indicating their amyloid structure (Fig 3C). Effective staining of the Vicilin fibrils by ThT (Fig 3D), a significant increase in the fluorescence quantum yield and lifetime of the bound dye (Fig 3E, right panels), and the presence of a pronounced long-wavelength shoulder in its absorption and fluorescence excitation spectra (Fig 3E, left panels) have confirmed the amyloid nature of these fibrils. It should be noted that alongside with the appearance of the long-wavelength shoulder in the absorption spectra of the bound dye, the concentration and, hence, the optical density of the bound ThT have increased (Fig 3E, dotted lines). This has led to the distortion of the short-wavelength region of the fluorescence spectra of ThT bound to Vicilin fibrils prepared by using the seeds. This is due to the secondary inner filter effect, which occurs when the absorption and fluorescence spectra of the sample substantially overlap and manifests in the reabsorption of light emitted by the sample. While determining the fluorescence quantum yield of the bound to fibrils ThT, this effect has been corrected by the usage of a specially developed experimental procedure [37]. Considering the obtained results, we have concluded that fibrillogenesis of the full-length Vicilin can be efficiently induced by the "seeds" of preincubated Vicilin or Cupin-1.2 aggregates.

Finally, to verify the amyloid properties of the Vicilin, Cupin-1.1, and Cupin-1.2 fibrils by the direct method, we obtained their x-ray diffraction patterns. All the samples for x-ray diffraction analysis have been prepared in the same conditions and have undergone the dialysis for desalting. The x-ray diffraction analysis revealed circular diffuse x-ray reflections 4.7 Å and 10 Å for the Vicilin fibrils (Fig 3F), circular x-ray reflections 4.7 to 4.88 Å and 8.6 Å for Cupin-1.1 fibrils, 4.7 to 4.9 Å and 8.8 Å for Cupin-1.2 fibrils (Fig 3F). These reflections are typical for bona fide amyloid fibrils with the cross-β sheet quaternary structure [50–54]. The 4.7 to 4.9 Å reflection is assumed to arise from the periodicity of the hydrogen-bonded β-strands oriented near perpendicular to the fiber axis, and the diffraction in the region of approximately 8 to 11 Å is presumed to relate to the stacking of these sheets parallel to the fiber axis [53,55,56]. It should be noted that, in contrast to the full-length Vicilin, both Cupin-1.1 and Cupin-1.2 samples exhibited additional reflections (Fig 3F) indicating presence of salt crystals that had not been removed by dialysis probably due to salt binding to the fibrils of Cupins.

Taken together, we demonstrate that full-length Vicilin and both its domains Cupin-1.1 and Cupin-1.2 under certain conditions form bona fide amyloid fibrils in vitro. Whereas Cupin-1.1 and Cupin-1.2 domains form amyloid fibrils after a simple incubation, amyloid

fibril formation by Vicilin is induced by the addition of "seeds" consisting of preformed aggregates of Vicilin or Cupin-1.2 but not Cupin-1.1.

### Vicilin, Cupin-1.1, and Cupin-1.2 exhibit amyloid properties being secreted to the surface of *Escherichia coli* cells and aggregate in *Saccharomyces cerevisiae*

We have tested aggregation properties of Vicilin and its Cupin domains heterologously expressed in vivo. Firstly, we have analyzed the fibrillation of the Vicilin, Cupin-1.1, and Cupin-1.2 when these proteins are secreted to the surface of the *E. coli* cells in the Curli-Dependent Amyloid Generator (C-DAG) system [57]. We have found that the secretion of Vicilin and Cupin-1.1 has caused a vivid orange-red color of bacterial colonies on the plates with CR, suggesting the extracellular aggregate formation by these proteins (Fig 4A). TEM has asserted this possibility demonstrating fibril formation by these proteins (Fig 4B). Notably, Cupin-1.2 has formed curved fibrils, whereas fibrils of Vicilin and Cupin-1.1 have been straighter (Fig 4B), correlating with a brighter color of the colonies secreting Vicilin and Cupin-1.1 and pale color of Cupin-1.2 colonies on the CR plates (Fig 4A). The polarization microscopy study of bacterial colonies secreting Vicilin, Cupin-1.1, and Cupin-1.2 fibrils has

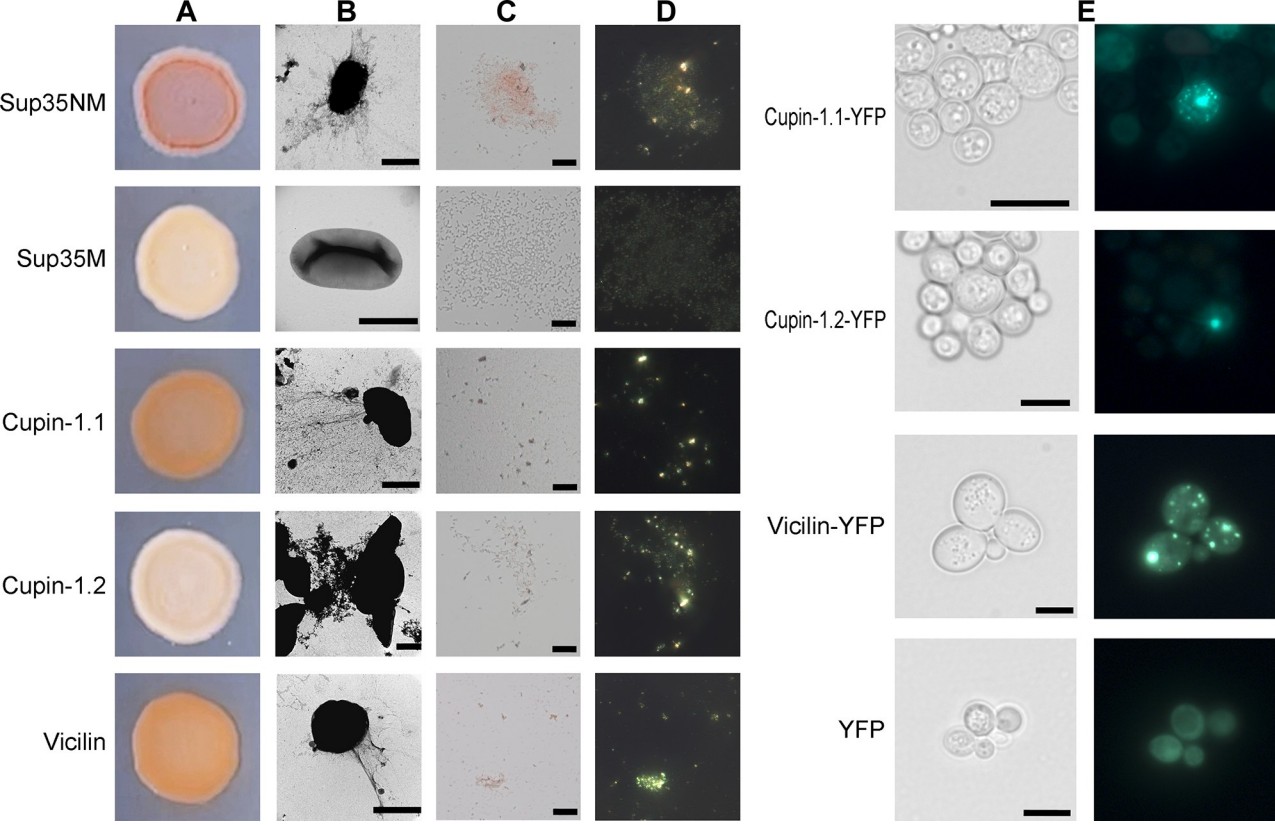

**Fig 4. Amyloid properties of Vicilin and its Cupin domains produced in bacteria and yeast cells.** (A) CR plate with *E. coli* cells secreting Vicilin, Cupin-1.1, and Cupin-1.2. The cells secreting either Sup35NM (amyloid) or Sup35M (soluble) proteins were used as the positive and negative controls, respectively. (B) TEM images of the Vicilin, Cupin-1.1, and Cupin-1.2 fibrils secreted by *E. coli* cells. Scale bars are equal to 500 nm (all images, with the exception of Cupin-1.2) or 1 μm (Cupin-1.2). (C and D) Vicilin, Cupin-1.1, and Cupin-1.2 fibrils secreted by *E. coli* cells bind CR and exhibit birefringence in polarized light. Transmitted light (C) and polarized light (D) images are shown, magnification 800× was used. Scale bars are equal to 20 μm. (E) The Vicilin, Cupin-1.1, and Cupin-1.2 proteins fused with YFP aggregate in the *S. cerevisiae* cells. Transmitted light (left) and fluorescent light (right) images are shown. Scale bars are equal to 5 μm. CR, Congo Red; TEM, transmission electron microscopy; YFP, yellow fluorescent protein.

revealed a strong birefringence of all 3 variants confirming their amyloid properties (Fig 4C and 4D).

In addition to the extracellular secretion of Vicilin, Cupin-1.1, and Cupin-1.2 in *E. coli*, we have overproduced these proteins fused with yellow fluorescent protein (YFP) in yeast *S. cerevisiae* to test its intracellular aggregation. The results of this experiment have demonstrated that all 3 proteins, Vicilin, Cupin-1.1, and Cupin-1.2, fused with YFP have formed fluorescent aggregates in yeast cells (Fig 4E). Thus, we have found that full-length Vicilin and both its Cupin-1 domains demonstrate amyloid properties not only in vitro but being heterologously expressed in vivo.

## Vicilin forms amyloid aggregates in mature seeds in vivo, and these aggregates resist canning and digestion by gastrointestinal enzymes

The most intriguing question was whether Vicilin forms amyloid aggregates in pea seeds in vivo. At the previous steps of this work, we demonstrated the formation of detergent-resistant Vicilin polymers in pea seeds by proteomic PSIA-LC-MALDI assay and its amyloid properties in vitro and heterologously expressed in vivo. To understand whether amyloid form of Vicilin can be detected in seeds histologically, we have analyzed the colocalization of Vicilin with amyloid-specific dye ThT on the cryosections of pea seed cotyledons (we used 30 days after pollination seeds where the accumulation of detergent-resistant form of Vicilin was detected by proteomic assay, S1 Table). Immunofluorescent microscopy carried on unfixed cryosections has shown an almost total overlapping of the anti-Vicilin antibody and ThT signals indicating the presence of the Vicilin amyloid aggregates (Fig 5A–5C; S2A–S2D Fig; S2 and S3 Data). The anti-Vicilin antibody signal has been localized predominantly in the central vacuole where it has demonstrated strong colocalization with ThT (Pearson's R value for pixels above threshold was 0.87, Costes *P* value was 1.0) and has revealed the presence of little granular compartments that have been proposed to be protein bodies (PB), the membrane compartments where Vicilin typically accumulates [58].

To confirm this observation, we have isolated PB from 30 days mature pea seeds using sucrose cushion sedimentation assay (see "Materials and methods") and have analyzed their staining with anti-Vicilin antibody and ThT. We have found that unfixed, air-dried PB exhibit total overlapping of the anti-Vicilin and ThT signals suggesting that the amyloid Vicilin aggregates are located in PB (laser scanning confocal microscopy, Fig 5D–5F; fluorescent microscopy, S2E–S2G Fig; single protein body, Fig 5G). Next, we have stained PB with CR and have demonstrated that PB are CR-positive and exhibit apple-green birefringence confirming their amyloid properties (Fig 5H). Scanning electron microscopy (SEM) of isolated PB revealed their near spherical morphology (Fig 5I). Immunoelectron microscopy with the polyclonal rabbit primary anti-Vicilin antibody and gold-labelled secondary anti-rabbit antibody has demonstrated weak antibody binding with untreated protein bodies (Fig 5J) due to presence of membrane that shields PB proteins from antibody binding. We have removed membranes using treatment of isolated PB with 1% Tween-20. This treatment has caused partial deformation of PB and has significantly increased antibody binding suggesting presence of Vicilin in PB (Fig 5K). Additional mild sonication of PB (5 s at 30% power, Q125 Qsonica, Newtown, CT) has caused a release of protein fibrils that have bound anti-Vicilin antibody (Fig 5L and 5M). The analysis of the protein content of PB by SDS-PAGE gel stained with Coomassie blue demonstrated that protein band corresponding to Vicilin was the most abundant in protein fraction of PB (Fig 5N, track Cm), and western blot analysis with the polyclonal anti-Vicilin antibody revealed presence of the Vicilin polymers partially dissolved by boiling in 2% SDS (Fig 5N, track B).

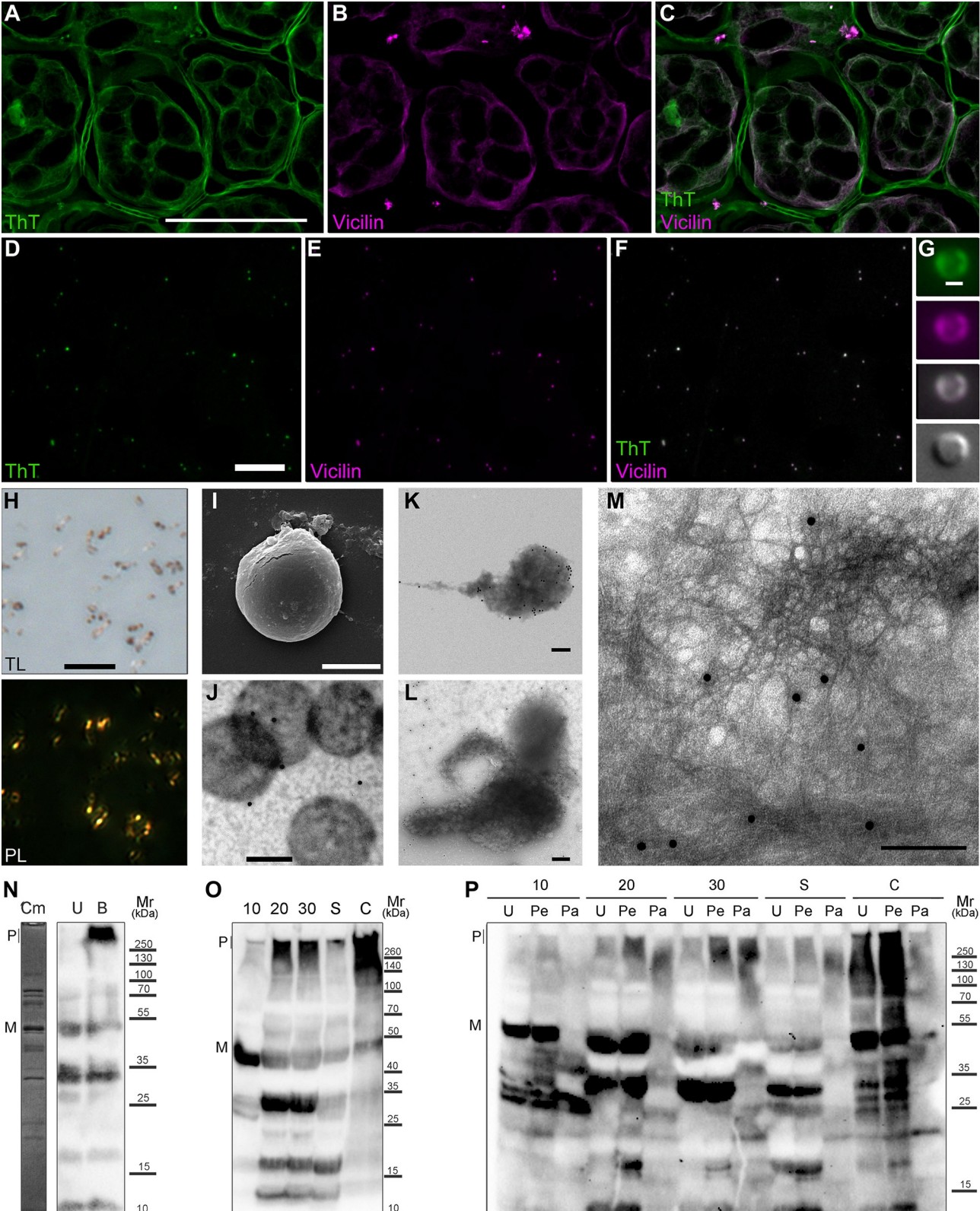

**Fig 5. Vicilin forms amyloid aggregates in vivo.** (A–C) In situ hybridization of anti-Vicilin antibodies with ThT on pea seed cryosections. (A) ThT channel of fluorescence (green), (B) anti-Vicilin antibody (magenta), (C) right—overlay (gray). Scale bar is equal to 100 μm. (D–F) In situ hybridization

of anti-Vicilin antibodies with ThT on PB extracted from pea seeds. Scale bar is equal to 50 μm. (G) Fluorescent microscopy of isolated protein body; channels from top to bottom: ThT, anti-Vicilin, overlay, differential interference contrast. Scale bar is equal to 2.5 μm. Source images are S2E–S2G Fig. (H) CR staining of isolated PB. TL, transmitted light; PL, polarized light. Scale bar is equal to 20 μm. (I) SEM of isolated protein body. Scale bar is equal to 2 μm. (J–M) Immunoelectron microscopy of isolated PB labelled with anti-Vicilin primary and gold-conjugated secondary antibodies: (J) without treatments, (K) treated with Tween-20, (L-M) treated with Tween-20 and sonicated. Scale bars are equal to 200 nm. (N) Vicilin content and its detergent resistance in the lyzates of PB. Cm: Coomassie-blue stained gel of the protein body lysate; U and B: western blot of the protein body lysate treated with 0.5% Tween-20 and 2% SDS, U, unboiled; B, boiled. (O) Detergent resistance of Vicilin aggregates isolated from pea seeds on different stages of maturation (10, 20, 30 days after pollination), germination (S, sprouts) and commercial canned peas (C, canned). (P) Protease resistance of Vicilin amyloids from pea seeds. U, untreated samples; Pe, treated with pepsin; Pa, treated with pepsin and then pancreatin; P, polymers; M, monomers. All samples in Panels O–P were loaded onto the gel after 5 min boiling in buffer with 2% SDS. Corresponding molecular weights are shown (kDa). CR, Congo Red; PB, protein bodies; SDS, sodium dodecyl sulfate; SEM, scanning electron microscopy; ThT, Thioflavin T.

Next, we have studied the accumulation of the amyloid, detergent-resistant Vicilin aggregates in pea seeds using total protein lysates obtained at the different stages of maturation or germination. Protein lysates were treated with 0.5% SDS and 0.5% Tween20 for 10 min at room temperature (RT), boiled for 10 min in the sample buffer containing 2% SDS (final concentration) and subject to SDS-PAGE followed by western blot with polyclonal anti-Vicilin antibody. The results of this experiment have confirmed the aforementioned proteomic (S1 Table) and histological (Fig 5A–5M) data and demonstrated that detergent-resistant aggregates of Vicilin tend to accumulate during the seed maturation (from 10 to 30 days after pollination) and reach their maximum in mature seeds (Fig 5O). Notably, these aggregates rapidly disassemble in germinating seeds (Fig 5O). Because aggregates of Vicilin are highly stable, we have decided to analyze their presence in the commercial canned peas produced by Bonduelle (Bonduelle Group, France) and Heinz (H.J. Heinz, Pittsburgh, PA). The results of this experiment have shown that Vicilin amyloid aggregates persist canning and retain in these food products (Fig 5O, S3 Fig).

We have analyzed the resistance of Vicilin amyloids to treatment with proteases. For this purpose, we have used in vitro protein digestibility assay (IVPD) that imitates gastrointestinal protein digestion [59]. Total protein lysates have been isolated from pea seeds, sprout cotyledons, and canned peas. Next, lysates have been consequently treated with pepsin and pancreatin, boiled in the sample buffer with 2% SDS, and subjected to SDS-PAGE and western blot with polyclonal anti-Vicilin antibody. Results of this experiment have demonstrated that in contrast to monomers, Vicilin amyloids from pea seeds resist proteolytic digestion demonstrating high stability (Fig 5P). Notably, in vitro obtained Vicilin fibrils were not resistant to pepsin and pancreatin (S4A Fig); nevertheless, they demonstrated resistance to trypsin treatment (S5 Fig), prolonged boiling with 2% SDS for 1 h (S4B Fig) and were efficiently dissolved only by concentrated formic acid (S6 Fig).

Taken together, we have demonstrated that Vicilin amyloids are present in pea seeds in vivo accumulating during the seed maturation, disassembling after germination, and retaining in food products like canned peas.

## Vicilin amyloid fibrils exhibit toxicity for yeast and mammalian cells

Because we have found Vicilin forms amyloids both in vivo and in vitro, we have decided to analyze the effects of Vicilin amyloid formation. Vicilins are known to have carbohydrate-binding lectin activity resulting in their toxicity for fungi and exhibit functional dualism being not only storage but also pathogen-defense proteins [60]. The present study has revealed that Vicilin can form at least 2 types of amyloid aggregates in vitro: less structured ones formed at the initial point of incubation and fibrils (Figs 1–3). To discriminate their effects, we tested toxicity of Vicilin, Cupin-1.1, and Cupin-1.2 nonfibrillar aggregates and fibrils for yeast cells at the same concentrations. The data obtained have demonstrated that Vicilin fibrils were very

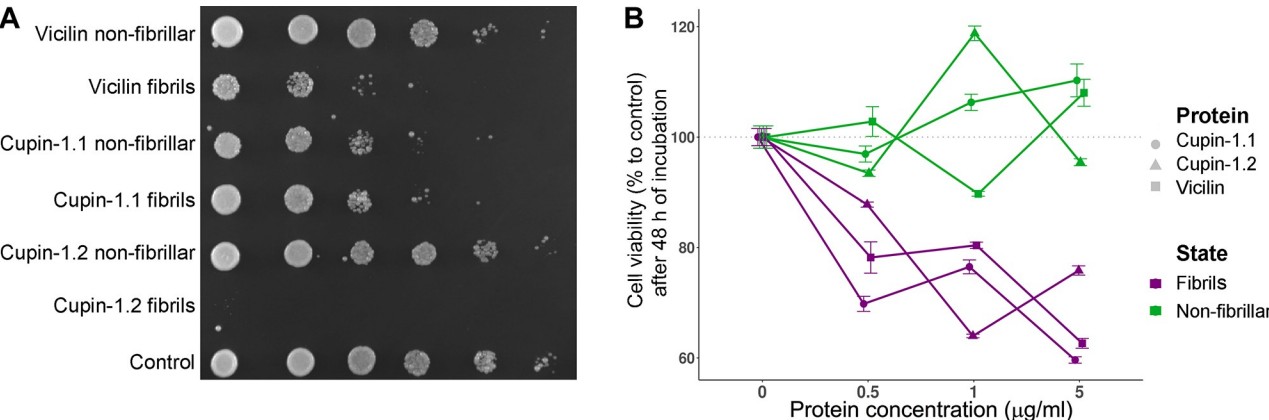

**Fig 6. Vicilin fibrils are toxic for yeast and mammalian cells.** (A) Analysis of the Vicilin, Cupin-1.1, and Cupin-1.2 fibrils and nonfibrillar aggregates toxicity for yeast cells. A series of 10-fold dilutions of the liquid yeast culture is shown. Picture was taken after 48 h of incubation at 30˚C. (B) Analysis of the Vicilin, Cupin-1.1, and Cupin-1.2 fibrils and nonfibrillar aggregates toxicity for mammalian cells. Dependence of the cell viability on the aggregates concentration is shown. Data were obtained after 48 h of incubation. Protein concentrations (μg/mL) are indicated. Scale bars correspond to the standard errors of the means. Values for the graph in Panel B are listed in S4 Data.

toxic for yeast culture resulting in significant yeast cells growth reduction after 48 h incubation (Fig 6A). Cupin-1.2 fibrils were extremely toxic causing almost complete death of cells (Fig 6A). In contrast to fibrils, nonfibrillar aggregates of Vicilin and Cupin-1.2 did not exhibit toxicity, whereas in the case of Cupin-1.1 both nonfibrillar aggregates and fibrils were toxic (Fig 6A). To confirm association of the Vicilin toxicity with its fibrillar amyloid state, we have sonicated the Vicilin fibrils to disrupt them (40s, at 60% power, Q125 Qsonica, Newtown, CT). Indeed, such treatment has caused significant reduction in the Vicilin, Cupin-1.1, and Cupin-1.2 toxicity for yeast cells (S7 Fig).

To evaluate potential toxicity of Vicilin amyloids towards mammalian cells, we exposed human colorectal adenocarcinoma DLD1 cells to nonfibrillar aggregates and fibrils of Vicilin, Cupin-1.1, and Cupin-1.2. We did not observe any toxicity when DLD1 cells were treated with nonfibrillar aggregates of Vicilin, Cupin-1.1, and Cupin-1.2 (Fig 6B). On the contrary, Vicilin, Cupin-1.1, and Cupin-1.2 fibrils decreased viability of the cells in concentration dependent mode (Fig 6B).

Overall, Vicilin toxicity for fungal and mammalian cells depends on its structural characteristics and significantly increases when forming amyloid fibrils. Both, Cupin-1.1 and Cupin-1.2 contribute to the toxicity but Cupin-1.2 fibrils exhibit especially high toxic effects for yeast cells.

## Discussion

Plant seed maturation and transition to dormancy are accompanied by the desiccation that poses a stressful condition to all the cell components. Genome stability during desiccation is maintained via expression of genes related to the DNA repair and chromatin remodeling [61,62] as well as chromatin compaction [63]. In contrast to the regulation of genome stability in seeds, molecular mechanisms underlying preservation of the seed proteins are poorly studied. Here, for the first time, we have obtained an experimental proof that full-length plant seed storage 7S globulin Vicilin exhibits amyloid properties in vivo and in vitro (Figs 1–5). Amyloid aggregates of Vicilin accumulate in PB during seed maturation and disassemble upon germination (Fig 5). These aggregates are protease- and detergent-resistant, bind amyloid-specific dyes ThT and CR, and have fibrillar morphology indicating their amyloid nature (Figs 1–5).

Thus, both genetic material of seeds and seed proteins undergo structural conversions aimed at compacting them to provide long-term storage and to overcome adverse conditions. Notably, plant seed amyloids could be presented not only by Vicilin but by ensembles of co-aggregating or interacting storage proteins because detergent-resistant aggregates of other major storage proteins, convicilins and legumins [64,65], were also detected in pea seeds by proteomic assay PSIA-LC-MALDI as well as important drought stress response protein Dehydrin [66] (S1 Table). Strong ThT fluorescence of the plant cell walls (Fig 5) also suggests the presence of amyloid aggregates in their structure as it was described for fungal cell walls [67,68].

Vicilin amyloids are efficiently solubilized only by concentrated formic acid being biochemically similar to extremely stable pathological human huntingtin exon-1 amyloids [69]. Such stability of the full-length protein is likely to be provided by the presence of 2 β-barrel domains Cupin-1.1 and -1.2 because they also form bona fide amyloid fibrils (Figs 1–4). Though Cupin-1.1 and Cupin-1.2 belong to the Cupin β-barrel domain superfamily, they have only 21.8% of identity. Nevertheless, both are predicted to be β-sheet-rich and contain 5 and 4 amyloidogenic regions, respectively (Fig 1A). Interestingly, amyloidogenic regions in the Cupin-1.2 domain are tightly stacked, whereas in Cupin-1.1 they are distributed over all parts of the domain (Fig 1A, S1 Data). Moreover, Cupin-1.2 has more charged and acidic amino acid composition and lower isoelectric point than Cupin-1.1, whereas the latter is more similar in these parameters to the full-length Vicilin. Such a difference in the amino acid sequences does not result in different physicochemical characteristics of the Cupin-1.1 and Cupin-1.2 fibrils but gives them some specific features including (i) the ability of Cupin-1.2 to form amyloid fibrils without protein dissolution in organic solvent (Fig 1), (ii) higher toxicity of the Cupin-1.2 fibrils in comparison with Cupin-1.1 fibrils (Fig 6), and (iii) the ability of the Cupin-1.2 fibrils to efficiently seed Vicilin fibrillation (Fig 3). Overall, based on these observations, Cupin-1.2 is likely to be the main structural determinant of the Vicilin amyloid formation.

Notably, being structurally similar to Cupins, β-barrel domain-containing proteins that belong to the outer membrane proteins (Omps) of Gram-negative bacteria also form amyloids [34,70,71] whereas human Aβ (1–42 aa) vice versa forms β-barrel pores in specific conditions [72]. In addition, recently, formation of β-barrel oligomer intermediates has been proposed as a common step in general mechanism of amyloid formation [73]. Thus, different eukaryotic and prokaryotic proteins containing β-barrel domains are amyloid-forming, and these domains are likely to be important and yet underestimated amyloidogenic determinants.

Formation of amyloids by Vicilin in vivo corresponds to the fact that plants should develop specific mechanisms to rescue nutrient reservoir in seeds for embryos during long-term dehydration and unfavorable conditions. A series of studies of the seed storage proteins in various plants species has demonstrated that most of them, including 7S Globulins, which Vicilin belongs to, form tri- or hexamers in vivo [74], whereas isolated seed protein mixtures or solitary seed proteins tend to form amyloid-like fibrils in vitro after limited hydrolysis, at high temperatures or extremal pH [20]. Such amyloidogenic nature of seed storage globulins containing ancient β-barrel domains of Cupin superfamily is predicted bioinformatically in most land plant species suggesting for high evolutionary conservation of this feature [22]. Amyloids are probably the most stable protein structures that resist different treatments and can persist in external environment for decades [75]. Thus, amyloid formation seems to be reasonable as evolutionary adaptation to provide a long-term survival of plant seeds. Moreover, similar examples of amyloids involved in protein storage were found in other kingdoms of life. For instance, egg envelope proteins of annual fish *Austrofundulus limnaeus* form amyloids during seasonal drought thus preventing dehydration [76], and human peptide hormones accumulate as protein granules with amyloid structure that disintegrate and release functional monomeric

hormones after pH change [77]. Thus, the storage function of amyloids is conservative not only in plants but exhibits cross-kingdom conservation between plants and animals. Significant decrease in the amounts of Vicilin amyloids, almost complete absence of the full-length Vicilin monomers and presence of proteolytic bands in germinating seeds (Fig 5) suggest that Vicilin amyloids are disassembled proteolytically, probably by serine and cystein proteases [78], rather than by chaperone machinery.

Not only storage but also defense from pathogens (primarily, fungi and insects) functions are typical for Vicilins [79] and associated with lectin (carbohydrate-binding) properties of these proteins [80,81] and their oligomerization [82,83]. Several lectins are known to form amyloid-like fibrils in vitro affecting their activity [84,85]. We have found out that the formation of amyloid fibrils significantly increases the toxicity of the full-length Vicilin for yeast in comparison with its unstructured aggregates (Fig 6), presupposing the role of amyloid formation in the defense function of this protein. Another property of lectins, in particular, Vicilins, is their allergenicity. Vicilin is one of the major plant-derived allergens, and its allergenic properties were found to be associated with its protease-resistant fragments [86,87]. Our finding that Vicilin forms amyloids in vivo explains its protease resistance (Fig 5) and suggests that these amyloids may represent major source of food allergy according to the data that in vitro generated lectin amyloids are phagocytized by macrophages and elicit an immune response [85]. Higher resistance of Vicilin amyloids extracted from pea seeds to protease treatment in comparison with the in vitro generated fibrils could be explained by their posttranslational modifications [88,89] or interaction with other molecules, for example, polysaccharides that could potentially ensure additional stabilization of Vicilin amyloids in vivo as it occurs with Aβ in human brain [90]. Higher amounts of Vicilin amyloids in the samples obtained from canned peas in comparison with fresh peas (Fig 5O and 5P) could be due to their thermal treatment during canning [91] that probably partially disassembles large amyloid aggregates and reduces their size thus allowing to enter stacking gel more efficiently.

Vicilin fibrils in high concentrations are toxic for mammalian cells (Fig 6) but concentrations of Vicilin amyloids in food products are many times lower; thus, their toxic effect for humans seems very unlikely. Nevertheless, the presence of Vicilin amyloids in canned peas (Fig 5) can potentially decrease food quality and increase allergic properties of seeds due to incomplete gastrointestinal digestion. Germination significantly reduces the amounts of amyloids (Fig 5); therefore, it is recommended to soak the seeds before eating to induce germination. The content of amyloids in plant seeds is likely to depend on the genetic variations in corresponding genes underlying structural properties of their protein products in different plant species and varieties. The creation of novel plant varieties with decreased amyloid formation of storage proteins may represent a promising strategy for future agriculture to improve nutritional value and reduce allergenicity of plant seeds.

Overall, in this study we have identified plant protein that forms amyloids under native conditions in vivo, have found that these amyloids mediate protein storage in plant seeds, and have demonstrated dynamics of their accumulation and disassembling, high stability, resistance to proteolytic digestion and canning as well as toxicity for fungal and mammalian cells.

## Materials and methods

### *P. sativum* L. plant material and growing conditions

The pea (*P. sativum* L.) line Sprint-2 from the collection of All-Russia Research Institute for Agricultural Microbiology (St. Petersburg, Russia) with determinate growth, early flowering start, and early seed maturation was used [92,93]. To obtain seeds at various stages of maturation, pea seeds were sown in pots with "Terra vita Universal" peat soil (MNPP FART, Russia)

with the following characteristics: pH (KCl) 6.0; 150 mg/L available nitrogen ($NH_4^+ + NO_3^-$); 270 mg/L available phosphorous ($P_2O_5$); and 300 mg/L available potassium ($K_2O$). Plants were grown in a constant environment chamber (model VB 1514, Vötsch, Germany) at 16/8 h and 24/22˚C day/night regime, 75% relative humidity, and around 10,000 lux illumination. The date of full opening of the flower was considered the date of pollination. Seeds intended for isolation of detergent-resistant protein fractions were collected 10, 20, and 30 days after pollination, which corresponded to the following stages of seed maturation: (i) flat pod, (ii) pod fill (green seeds), and (iii) yellow wrinkled pod [94]. Several plants cultivated were left until the dry harvest stage [94] in order to study the effect of germination on the seed detergent-resistant fractions content afterward. Harvested dry pea seeds were surface-disinfected for 6 min in 98% sulfuric acid, then thoroughly rinsed with sterile water and placed on Petri dishes containing sterile 1% agar-agar. After the seed germinating for 3 days at 27˚C in the dark seedlings without signs of microbial contamination were selected and their cotyledons were cut off. Plant seeds and cotyledons collected were immediately frozen in liquid nitrogen and then stored at −80˚C.

## Microbial strains, plasmids, and cultivation conditions

The plasmids for analysis of aggregation of Vicilin, Cupin-1.1, and Cupin-1.2 fused with YFP in yeast *S. cerevisiae* were obtained by the insertion of PCR-amplified fragments of the corresponding genes amplified with primer pairs containing inserted *Bam*HI (reverse) and *Hind*III (forward) restriction sites (S3 Table) and pea seed cDNA fragment as template into the pRS315-CUP1-SIS1-YFP plasmid [95] by the *Bam*HI and *Hind*III sites, respectively. Total RNA from pea seeds was extracted using Trizol (Invitrogen, Carlsbad, CA) and cDNA was prepared with SuperScript III reverse transcriptase (Invitrogen, Carlsbad, CA). *E. coli* strain DH5α [96] was used for plasmid amplification. The insertions of the corresponding genes were confirmed by the sequencing by using the CUP1 primer [95].

The *S. cerevisiae* 1-OT56 strain (*MAT*a *ade1-14*$_{UGA}$ *his3 leu2 trp1-289*$_{UAG}$ *ura3* [*psi⁻*] [*PIN⁺*]) [97] was transformed with constructed plasmids. Yeast cultivation was performed on selective agar media at 30˚C for 4 days followed by replacing on agar media with 150 μL ml$^{-1}$ of $CuSO_4$ for *CUP1* promoter induction. Fluorescence was analyzed using a Zeiss Axio Imager A2 fluorescent microscope (Carl Zeiss, Germany).

To construct plasmids for Vicilin, Cupin-1.1, and Cupin-1.2 export by C-DAG system the fragments of interest were amplified by PCR using pairs of primers flanked with *Not*1 (forward) and *Sal*1 (reverse) restriction sites (S3 Table) and pea seed cDNA as template. The PCR products and pExport vector [57] were digested by *Not*1 and *Sal*1 restriction enzymes and then ligated. Plasmids encoding yeast Sup35NM (aa 2–253) and Sup35M (aa 125–253) fused with CsgA signal sequence were constructed previously [57].

## C-DAG assay

Analysis of amyloid properties of Vicilin, Cupin-1.1, and Cupin-1.2 with the usage of C-DAG system was performed as described earlier [57]. To export proteins on the cell surface *E. coli* strain VS39 [57] was transformed with pExport-based plasmids, encoding Vicilin, Cupin-1.1, and Cupin-1.2 fused with CsgA signal sequence. Export of amyloid-forming Sup35NM and nonamyloidogenic Sup35M proteins was used as positive and negative control of amyloid formation, respectively. Birefringence analysis was performed with a usage of Axio Imager A2 transmitted light microscope (Carl Zeiss, Germany) equipped with a 40× objective and cross-polarizers. For TEM analysis incubation on inducing plates without CR dye was used.

## Mammalian cell lines and toxicity assay

Human DLD1 cells were seeded at $5 \times 10^4$ cells per well density on 24-well plates and exposed to protein samples analyzed. At indicated time points, medium was replaced with 0.5 mg/mL MTT solution for 1 h at 37°C. Then formazan was dissolved in DMSO, and optical densities were measured at 572 nm wavelength using the Multiscan Ex spectrophotometer (Thermo Fisher Scientific, Waltham, MA). Data were presented as the mean of 3 independent experiments ± the standard error of the mean.

## PSIA-LC-MALDI proteomic assay

The PSIA-LC-MALDI proteomic assay for the identification of protein complexes resistant to the treatment with ionic detergents has been published previously [27]. Here, we have used this protocol with several modifications. Frozen pea seed cotyledons (1 g) were disrupted with pestle and mortar in liquid nitrogen. Resulting powder was dissolved in 5 mL of PBS buffer and treated with 1% SDS and 0.2% Tween-20 detergents for 15 min. Next, lysate was sedimented at 10,000 g for 5 min; supernatant was moved to a new tube and sedimented again. Resulting supernatant was loaded onto the top of 1.5 ml of 25% sucrose/PBS with 0.1% SDS cushion and centrifuged for 8 h, 225,000$g$ at 18°C in L8-70 ultracentrifuge (Beckman Coulter, Brea, CA), Type50 Ti angle rotor (Beckman Coulter, Brea, CA). Resulting pellets were washed in 5 mL of distilled water and centrifugated again (1.5 h, 225,000$g$, 4°C, L8-70 ultracentrifuge (Beckman Coulter, Brea, CA), Type50 Ti angle rotor (Beckman Coulter, Brea, CA). This step was repeated twice.

Next, samples were lyophilized, treated with formic acid to solubilize proteins, and lyophilized again. Salts and detergents were removed, and samples were subjected to trypsinolysis stage followed by reverse phase high performance liquid chromatography and mass spectrometry as described earlier [27]. During analysis, preset parameters of "Mass tolerance" were used (precursor mass tolerance 50 ppm, fragment mass tolerance 0.9 Da). Peptide Calibration Standard II 8222570 (Bruker Daltonics, Billerica, MA) was applied as a standard sample. Carboxymethylation of cysteine, partial oxidation of methionine, and one skipped trypsinolysis site were considered as permissible modifications. The obtained mass spectra were matched to the corresponding proteins using NCBI database. The BioTools software (Bruker Daltonics, Billerica, MA) was used for manual validation of protein identification.

## Protein expression and purification

To express the Vicilin, Cupin-1.1, and Cupin-1.2 C-terminally fused with a 6x-His tag, the Alicator kit (Thermo Fisher Scientific, Waltham, MA) was used. The fragments were PCR amplified using respective primer pairs (S3 Table) and pea seed cDNA as template. The PCR-amplified fragments were inserted into the pAlicator vector (Thermo Fisher Scientific, Waltham, MA) according to the manufacturer's recommendations. The correctness of the resulting pAc-Vicilin, pAc-Cupin-1.1, and pAc-Cupin-1.2 plasmids was verified by sequencing with the primers provided by the manufacturer (Thermo Fisher Scientific, Waltham, MA).

For protein expression, *E. coli* strain BL21 (New England Biolabs, Ipswich, MA) was used. The overproduction of recombinant proteins was carried out in 2TYa media supplemented with 0.1 mM IPTG. Cultures were grown at 37°C for 4 h. Proteins were purified in denaturing conditions (in the presence of 8 M urea) according to a previously published protocol [98] without the Q-sepharose purification step. A one-step purification procedure with a Ni-NTA agarose (Invitrogen, Carlsbad, CA) column was performed according to the manufacturer's recommendations. Proteins were concentrated using ethanol instead of methanol used in the original protocol.

## In vitro protein fibrillation

For initiation of Vicilin and its domains aggregation in vitro, different buffers and incubation conditions were used. At the first stage the proteins in concentration 0.5 mg/mL were incubated in 5 mM potassium phosphate buffer [pH 7.4] at room temperature and at 50˚C with constant stirring for 2 weeks. Furthermore, the proteins in the same concentration were dissolved in 50% HFIP (Sigma-Aldrich, St. Louis, MO) and incubated for 7 days. Afterwards, the HFIP was slowly evaporated under a stream of nitrogen to one-third to one-fourth of sample volume; then the volume of the sample was adjusted with distilled water to the initial one, and the samples were stirred for an additional 7 days. This solvent (often, 100% HFIP) is widely used to ensure that proteins and peptides are entirely monomeric before starting analogical aggregation experiments. Moreover, a method was previously developed for the standardized and biocompatible preparation of aggregate-free Aβ for biophysical and biological studies of Alzheimer disease [99], one of the stages of which is the dissolution of the peptide in HFIP. Considering the fact that Vicilin, Cupin-1.1, and Cupin-1.2 are poorly soluble in ordinary buffers, we suggest that the use of HFIP with its subsequent removal from the sample after protein dissolving (that is, fibrillogenesis occurs in distilled water, not in HFIP) is the mildest of the possible effects minimally affecting the structure and properties of the protein. These conditions were also used for experiments with seeding. "Seeds" were prepared on the basis of the previously obtained Vicilin mixture of fibrillar and nonfibrillar aggregates, Cupin-1.1 fibrils, and Cupin-1.2 fibrils. These seeds were added to the samples at the beginning of fibrillogenesis in 1% (v/v) concentration.

## Pea seed protein extraction, in gel separation, and transfer

A standard protocol was used for total protein isolation from pea seeds. Three seeds collected at indicated stage of maturation or cotyledons of 3 seeds after germination were analyzed in each sample. Canned peas produced by Bonduelle (Bonduelle Group, France) and Heinz (H.J. Heinz, Pittsburgh, PA) were purchased in a supermarket in St. Petersburg, Russia. Homogenization of seeds was performed using glass beads. Protein concentrations were equilibrated by using Qubit 3 Fluorometer (Invitrogen, Carlsbad, CA). Then detergents were added sequentially to the final concentrations: 0.5% Tween20 (Helicon, Russia), 0.5% SDS (Helicon, Russia) followed by 10 min incubation at RT. Further, if necessary, IVPD assay was applied. Next, samples were boiled in the presence of 2% SDS for 5 min and loaded onto the 10% SDS-PAGE gel (Bio-Rad, Hercules, CA). After SDS-PAGE, wet transfer (Bio-Rad, Hercules, CA) was performed using PVDF membrane (GE Healthcare, Chicago, IL).

## IVPD

Pea seed proteins were digested in vitro as described previously [59]. After each step of enzyme treatment, samples were collected and inactivated by heating at 100˚C for 5 min. Pepsin (Roche, Germany) and pancreatin from porcine pancreas (Sigma-Aldrich, St. Louis, MO) were used. Samples were checked with SDS-PAGE followed by western blot with polyclonal anti-Vicilin antibody (Imtek, Russia). All analyses were performed in quadruplicate.

## Immunochemical analysis

To obtain polyclonal anti-Vicilin antibodies from serum of a healthy rabbit, immunization with the antigen of the recombinant full-length Vicilin obtained according to the protocol described above (without the step of fibrillation) was performed. Rabbit immunization and purification using affinity chromatography with rabbit antisera on a sorbent with immobilized

recombinant Vicilin protein were carried out in Imtek company (Imtek, Russia). Antibody was checked for recognition of the full-length Vicilin and its domains, Cupin-1.1 and Cupin-1.2, and for the absence of cross-reactivity with the *E. coli* proteins (S8 Fig). Dilution 1:1,000 was used. Further goat anti-rabbit IgG (H+L) secondary antibody (Thermo Fisher Scientific, Waltham, MA) was used in dilution 1:33,000. Also, 6x-His epitope tag antibody (4A12E4, Invitrogen, Carlsbad, CA) was used in dilution 1:5,000; then goat anti-mouse IgG (H+L) secondary antibody (Invitrogen, Carlsbad, CA) was used in dilution 1:5,000.

## Protein body isolation

PB were extracted from fresh pea seeds of 30 days after pollination according to modified protocol for *Zea mays* PB [100]. In particular, 0.5 g of the seeds were grounded with cooled mortar and pestle in the presence of 5 mL of 10% (w/w) sucrose in PBS containing 1 mM phenylmethylsulfonyl fluoride. Resulting homogenate was filtered through the nylon cloth and filtrate was then centrifuged at 500$g$ and 4°C for 10 min. Supernatant was layered onto discontinuous sucrose gradient comprising of 1, 1.5, and 1.5 mL of 70%, 50%, and 20% (w/w) sucrose cushions in PBS, respectively, in the open-top thickwall polycarbonate tube (Beckman Coulter, Brea, CA) and centrifuged in L8-70 ultracentrifuge (Beckman Coulter, Brea, CA) in Type50 Ti angle rotor (Beckman Coulter, Brea, CA) at 36,500$g$ and 4°C for 1 h. PB were collected from the boundary of 70% and 50% cushions as a part of opaque white halo. Approximately 100 μL of this fraction was extracted, 10-fold diluted with PBS, and centrifuged at 12,000$g$, 4°C, 5 min. The supernatant was discarded and the pellet was resuspended in 10% sucrose in PBS.

## In situ hybridization and fluorescent microscopy

Slices preparation: mature pea seeds were glued to holder with Tissue-Tek O.C.T. Compound (Sakura Finetek, Japan) and cut with a cryostat CM3050 S (Leica Biosystems, Germany) at −16°C to 20-μm-thick sections which were mounted on poly-l-lysine coated slides. To promote better adhesion, slides were kept for at least 12 h at 37°C. PB preparation: sucrose solution of PB was applied on a poly-l-lysine coated slides and air-dried for 30 min at room temperature. Slides were washed twice for 5 min in tris-buffered saline (TBS, 0.1 M), blocked for 2 h in a humid chamber at RT with solution containing 10% normal goat serum (Sigma-Aldrich, St. Louis, MO), 0.1% Triton X-100 (Sigma-Aldrich, St. Louis, MO) on TBS. Sections were incubated with primary antibodies in a humid chamber at 4°C overnight. Polyclonal rabbit-anti-Vicilin antibodies were applied at concentration 1:100 in TBS with 0.01% Triton X-100 and 1% normal goat serum. The primary antibodies were detected by 2 h incubation at 37°C with secondary antibodies solution: goat anti-rabbit IgG, conjugated to Alexa Fluor 568 (Thermo Fisher Scientific, Waltham, MA) diluted 1:250 in TBS containing 1% normal goat serum and 0.01% Triton X-100. Sections were washed twice with TBS. Slides were additionally stained for 40 min with 0.5% solution of ThT (Sigma-Aldrich, St. Louis, MO) in 0.1 N HCl, briefly differentiated in 70% ethanol, washed by TBS, and mounted in 80% TBS-glycerol. Longer staining times were avoided because of rapid washing-off of immunofluorescent label. A control for nonspecific staining was performed by replacing the primary antibody produced in rabbit with 10% normal rabbit serum. Sections were analyzed with the fluorescence microscope Leica DM5500 and scanning confocal microscope Leica TCS SP5 (Leica Biosystems, Germany).

## TEM

Micrographs were obtained using a transmission electron microscope Libra 120 (Carl Zeiss, Germany). The samples were placed on nickel grids coated with formvar films (Electron Microscopy Sciences, Hatfield, PA). To obtain electron micrographs, the method of negative

staining with a 1% aqueous solution of uranyl acetate was used. Immunoelectron microscopy was performed as described previously [34]. The rabbit polyclonal anti-Vicilin antibody at 1:50 dilution and goat anti-rabbit secondary IgG antibody conjugated with gold particle (Electron Microscopy Sciences, Hatfield, PA) at 1:40 dilution were used.

## SEM

Isolated protein bodies were dried on coverslips at RT. Coverslips were mounted on circular stubs with double-sided conductive tape to assure good electrical contact after coating. The preparations were coated with platinum to a thickness of 25 nm with Leica EM SCD500 (Leica Biosystems, Germany) sputter coater and examined with Tescan MIRA3 LMU microscope (Tescan, Czech Republic) at an acceleration potential of 9kV using in-beam SE detector.

## Polarization microscopy

CR staining of amyloid samples was performed using saturated CR (Sigma-Aldrich, St. Louis, MO) solution filtered through 45-μm filter (Merck Millipore, Burlington, MA). Samples stained with CR were air-dried and rigorously washed with distilled water. Birefringence was analyzed by using Zeiss Axio Imager A2 (Carl Zeiss, Germany) polarization microscope equipped with cross-polarizers.

## ThT staining and confocal microscopy

ThT UltraPure Grade (AnaSpec, Fremont, CA) without after-purification was used. ThT-fibrils–tested solutions were prepared by equilibrium microdialysis using a Harvard Apparatus/Amika device (Harvard Apparatus, Holliston, MA). Equilibrium microdialysis was performed with a concentration of aggregates approximately 0.5 mg/mL and initial concentration of ThT approximately 32 μM. Spectroscopic study of the sample and reference solutions prepared by the proposed approach allowed us to determine the photophysical characteristics of ThT bound to tested amyloids [36].

For obtaining the fluorescence images of the ThT-stained fibrillar structures confocal laser scanning microscope Olympus FV 3000 (Olympus, Japan) was used. We used the oil immersion objective with a 60× magnification, numerical aperture NA 1.42, and laser with excitation line 405 nm.

## Spectral measurements

The absorption spectra of the samples were recorded by using a U-3900H spectrophotometer (Hitachi, Japan). The absorption spectra of proteins aggregates and ThT in their presence were analyzed along with the light scattering using a standard procedure [101]. The concentrations of Vicilin, Cupin-1.1, Cupin-1.2, and ThT in solutions were determined using molar extinction coefficients of $\varepsilon_{280} = 47{,}271$ $M^{-1}cm^{-1}$, $\varepsilon_{280} = 16{,}928$ $M^{-1}cm^{-1}$, $\varepsilon_{280} = 18{,}549$ $M^{-1}cm^{-1}$, and $\varepsilon_{412} = 31{,}589$ $M^{-1}cm^{-1}$, respectively.

Fluorescence and fluorescence excitation spectra were measured by using a Cary Eclipse spectrofluorimeter (Varian, Australia). Fluorescence of ThT was excited at a wavelength of 440 nm and registered at 490 nm. A PBS solution of ATTO-425, whose fluorescence and absorption spectra are similar to that of ThT, was taken as a reference for determining the fluorescence quantum yield of ThT bound to fibrils. The fluorescence quantum yield of ATTO-425 was taken as 0.9 (ATTO-TEC Catalogue 2009/2010 p.14). The spectral slits width was 5 nm in most of experiments. Changing the slit widths did not influence the experimental results.

Recorded fluorescence intensity was corrected on the primary inner filter effect with the use of previously elaborated approach [37].

CD spectra in the far-UV region were measured by using a J-810 spectropolarimeter (Jasco, Japan). Spectra were recorded in a 0.1-cm cell from 260 to 200 nm. For all spectra, an average of 3 scans was obtained. The CD spectrum of the appropriate buffer was recorded and subtracted from the samples spectra. It has turned out that the recorded CD spectra of proteins aggregates have a clear distortion in the region of 250 to 260 nm (Fig 2A, right panel). According to literature, light scattering of macromolecules can substantially distort the CD spectra (see, for example, the work by Tinoco, Bustamante, and Maestre [102]). The light scattering of prepared amyloids, indeed, turned out to be very high in comparison to protein monomers (Fig 2A, Inset). We attempted a quantitative analysis of the secondary structure by the CDPro program using 3 different regression methods (Selcon, Contin, and CDSSTR) and several basic sets of proteins with a known secondary structure (the sets include from 37 to 56 soluble, membrane, and denatured proteins with different content of the secondary structure). Because such results could be arbitrary because the standard basic sets of proteins used for estimation of the secondary structure content are not representative for the analysis of the spectra of protein aggregates, we have also conducted a visual analysis of the recorded spectra with the use of CD spectra of proteins and peptides with representative secondary structures [39].

## Time-resolved fluorescence measurements

Fluorescence decay curves were recorded by a spectrometer FluoTime 300 (PicoQuant, Germany) with the Laser Diode Head LDH-C-440 ($\lambda_{ex}$ = 440 nm). The fluorescence of ThT was registered at $\lambda_{em}$ = 490 nm. Using recorded fluorescence decay fluorescence lifetime of ThT bound to studied aggregates was calculated. For this, the measured emission decays were fit to a multiexponential function using the standard convolute-and-compare nonlinear least-squares procedure [103]. In this method, the convolution of the model exponential function with the instrument response function was compared to the experimental data until a satisfactory fit was obtained. The fitting routine was based on the nonlinear least-squares method. Minimization was performed according to Marquardt [104].

## X-ray diffraction analysis

The full-length Vicilin, Cupin-1.1, and Cupin-1.2 fibrils for x-ray diffraction analysis were prepared in vitro using protein dissolving in 50% HFIP followed by HFIP evaporation and protein incubation for 7 days in distilled water. To obtain Vicilin fibrils, inoculation with preformed Vicilin "seeds" (1/100) was used. Additional washing of the obtained fibrils was carried out using equilibrium dialysis. Samples in dialysis tubing (Scienova, Germany) with a pore size of 12 to 14 kDa were placed into containers with filtered deionized water in the ratio sample/water = 1/500 (v/v). Water was changed after 2 h, then after next 3 h and, finally, after a night of incubation of the samples. Dialysis has been finished one day after the start of the incubation. The Vicilin, Cupin-1.1, and Cupin-1.2 fibrils (2 mL of 1 mg/mL solution) were lyophilized by using FreeZone 1L lyophilizator (Labconco, Kansas City, MO). The freeze-dried samples (2 mg) were dissolved in 20 μL of the miliQ water (to the final concentration approximately 100 mg/mL). Droplets of these preparations were placed between the ends of wax-coated glass capillaries (approximately 1 mm in diameter) separated approximately by 1.5 mm. The fibril diffraction images of Cupin-1.1 were collected by using a Bruker AXS X8 Microstar x-ray generator with HELIOX optics equipped with a Bruker AXS Platinum135 CCD detector (Bruker AXS, Madison, WI). Cu Kα radiation, λ = 1.54 Å (1 Å = 0.1 nm) was used. The samples were positioned at the right angle to the x-ray beam using a 4-axis kappa goniometer. The

diffraction images of the Vicilin and Cupin-1.2 fibrils were collected on a Rigaku XtaLab Synergy S instrument (Rigaku, Japan) with a HyPix detector and a PhotonJet microfocus x-ray tube by using Cu Kα (1.54184 Å) radiation. Images were prepared using the powder power tool in the CrysAlisPro data reduction package. Experiments were carried out with a 360˚ phi rotation and exposure time was 300 s.

## Bioinformatic predictions of protein properties

Physicochemical properties of Cupin-1.1 and Cupin-1.2 were calculated using pepstats program from EMBOSS package [28]. Comparison of the amino acid sequences of Cupin domains was performed using Smith-Waterman alignment with water program from EMBOSS package [28,29]. The monomeric structure of Vicilin was predicted by I-TASSER server [30]. Potentially amyloidogenic regions were predicted using AmylPred2 [31] and AmyloidMutants [32] methods.

## Data analysis

All experiments in this work were performed in at least 3 repeats. Data in figures are presented as the mean ± the standard error of the mean or ± the standard deviation. In human cell toxicity assay, a dependence of the toxicity on the concentration was considered as significant if the $p$-value of the regression coefficient in generalized linear model (glm function in R language) was lower than 0.05 and therefore the regression coefficient was significantly different from zero.

## Supporting information

**S1 Fig. Mass spectrometry identification data of the *P. sativum* Vicilin (47 kDa).** Amino acid sequence of Vicilin is shown. Peptides identified by mass spectrometry are indicated in red. Signal peptide is highlighted in green.
(PDF)

**S2 Fig. ThT colocalizes with anti-Vicilin antibody at pea seeds cryosections and extracted PB.** (A–D) Laser scanning confocal microscopy of the in situ hybridization of anti-Vicilin antibodies with ThT on pea seed cryosections is shown. ThT channel of fluorescence is green, anti-Vicilin antibody—red, overlay—yellow and gray. (A) ThT channel, yellow outline shows the region used for colocalization analysis of cytoplasm; (B) anti-Vicilin; (C) overlay of the anti-Vicilin and ThT channels, insert shows correlation diagram between channels; (D) anti-Vicilin and ThT channels, colocalized pixels shown in grayscale. (E–G) ThT staining of PB isolated from pea seeds. (E) In situ hybridization with anti-Vicilin antibodies. (F) ThT staining of the same slide. (G) Differential interference contrast. Scale bar is equal to 50 μm. PB, protein bodies; ThT, Thioflavin T.
(TIF)

**S3 Fig. Vicilin amyloids resist canning.** Western blot image is shown, polyclonal anti-Vicilin antibody was used. B: canned peas produced by Bonduelle (Bonduelle Group, France), H: the same by Heinz (H.J. Heinz, Pittsburgh, PA), P: polymers, M: monomers. All samples were loaded onto the gel after 5 min boiling in buffer with 2% SDS. Corresponding molecular weights are shown (kDa). SDS, Sodium Dodecyl Sulfate.
(TIF)

**S4 Fig. In vitro obtained Vicilin fibrils are not resistant to pancreatin and pepsin treatment but resist prolonged boiling with 2% SDS.** Western blot image is shown, polyclonal anti-

Vicilin antibody was used. (A) Protease resistance of the Vicilin, Cupin-1.1, and Cupin-1.2 in vitro obtained fibrils. U, untreated samples; Pe, treated with pepsin; Pa, treated with pepsin and pancreatin; P, polymers; M, monomers. All samples in sections g–h were loaded onto the gel after 5 min boiling in buffer with 2% SDS. Corresponding molecular weights are shown (kDa). (B) Vicilin (V), Cupin-1.1 (C1), and Cupin-1.2 (C2) fibrils obtained in vitro were boiled for 1 h in buffer contained 2% SDS and then loaded onto the gel. Corresponding molecular weights are shown (kDa). SDS, Sodium Dodecyl Sulfate.
(TIF)

**S5 Fig. In vitro obtained Vicilin fibrils are resistant to trypsin digestion.** Western blot image is shown; polyclonal anti-Vicilin antibody was used. U, untreated samples; Tr, treated with trypsin at 1:60 trypsin-to-total protein mass ratio for 45 min at 37°C; Pe, treated with pepsin as described in "Materials and methods." Samples were loaded onto the gel after 5 min boiling in buffer with 2% SDS. Corresponding molecular weights are shown (kDa). SDS, Sodium Dodecyl Sulfate.
(TIF)

**S6 Fig. In vitro obtained Vicilin fibrils are dissolved by formic acid.** Western blot image is shown, polyclonal anti-Vicilin antibody was used. U, samples treated with cold 2% SDS; B, samples boiled in buffer containing 2% SDS; F, lyophilized samples were treated with 90% formic acid (Sigma-Aldrich, St. Louis, MO), lyophilized again, solubilized in the same volume of PBS with addition of SDS sample buffer containing 2% SDS (final concentration) and loaded onto the gel. Corresponding molecular weights are shown (kDa). SDS, Sodium Dodecyl Sulfate.
(TIF)

**S7 Fig. Sonication decreases toxicity of the Vicilin, Cupin-1.1, and Cupin-1.2 fibrils.** (A) A series of 10-fold dilutions of the liquid yeast culture treated with fibrillar, nonfibrillar, and disrupted by sonication aggregates of the corresponding proteins is shown. 1 mg/mL protein concentrations were used. Sonicated samples were prepared from fibrils using the following conditions: 40 s, at 60% power, Q125 sonicator (Qsonica, Newtown, CT). Picture was taken after 48 h of incubation at 30°C. (B) The effect of sonication on the fibrils of Vicilin, Cupin-1.1, and Cupin-1.2. The same sonication conditions as in Panel A have been used. Samples have been stained with CR. TL, transmitted light; PL, polarized light. The scale bar is equal to 20 μm. CR, Congo Red.
(TIF)

**S8 Fig. Analysis of the polyclonal rabbit anti-Vicilin antibody specificity.** Left: Coomassie blue stained SDS-PAGE gel; right: western blot image. C1, recombinant Cupin-1.1 protein sample; C2, recombinant Cupin-1.2 sample; V, recombinant full-length Vicilin sample; E, total protein lysate of *E. coli* strain BL-21. All samples were boiled in buffer containing 2% SDS (final concentration) and loaded onto the gel. Corresponding molecular weights are shown (kDa). SDS, Sodium Dodecyl Sulfate.
(TIF)

**S1 Table. Proteins identified in detergent-resistant fraction of the *P. sativum* L. seeds.**
(PDF)

**S2 Table. Content of the secondary structure elements (%) in plant proteins and human Aβ (1–42 aa).** Values for S2 Table are presented in the supporting information (S4 Data).
(PDF)

**S3 Table. Oligonucleotides used in this study.**
(PDF)

**S1 Data. Per-residue β-sheet propensity for the full-length Vicilin protein and its domains, Cupin-1.1 and Cupin-1.2.**
(PDF)

**S2 Data. Colocalization analysis of the ThT and anti-Vicilin antibody at the pea seed cryosections in Coloc2 plugin, full field of view.** ThT, Thioflavin T.
(PDF)

**S3 Data. Colocalization analysis of the ThT and anti-Vicilin antibody at the pea seed cryosections in Coloc2 plugin, cytoplasm only.** ThT, Thioflavin T.
(PDF)

**S4 Data. Values for all data used to create the graphs throughout the paper.**
(XLSX)

**S1 Raw Images. Uncropped blots shown throughout the paper.**
(PDF)

## Acknowledgments

The authors thank Dr. A. A. Rubel (St. Petersburg State University) for kindly providing Aβ (1–42 aa). The authors acknowledge Collective Use Center for Genome Technologies, Proteomics, and Cell Biology (All-Russia Research Institute of Agricultiral Microbiology) and Center for Molecular and Cell Technologies (Research Park, St. Petersburg State University) for the equipment provided for use in this work.

## Author Contributions

**Conceptualization:** Kirill S. Antonets, Anton A. Nizhnikov.

**Formal analysis:** Kirill S. Antonets, Mikhail V. Belousov, Anna I. Sulatskaya, Maria E. Belousova, Anastasiia O. Kosolapova, Maksim I. Sulatsky, Elena A. Andreeva, Pavel A. Zykin, Yury V. Malovichko, Oksana Y. Shtark, Anna N. Lykholay, Kirill V. Volkov, Irina M. Kuznetsova, Konstantin K. Turoverov, Elena Y. Kochetkova, Alexander G. Bobylev, Konstantin S. Usachev, Oleg. N. Demidov, Igor A. Tikhonovich.

**Funding acquisition:** Anton A. Nizhnikov.

**Investigation:** Kirill S. Antonets, Mikhail V. Belousov, Anna I. Sulatskaya, Maria E. Belousova, Anastasiia O. Kosolapova, Maksim I. Sulatsky, Elena A. Andreeva, Pavel A. Zykin, Yury V. Malovichko, Oksana Y. Shtark, Anna N. Lykholay, Kirill V. Volkov, Elena Y. Kochetkova, Alexander G. Bobylev, Konstantin S. Usachev, Oleg. N. Demidov, Anton A. Nizhnikov.

**Methodology:** Kirill S. Antonets, Mikhail V. Belousov, Anna I. Sulatskaya, Maksim I. Sulatsky, Pavel A. Zykin, Yury V. Malovichko, Oksana Y. Shtark, Anna N. Lykholay, Kirill V. Volkov, Alexander G. Bobylev, Konstantin S. Usachev, Oleg. N. Demidov, Anton A. Nizhnikov.

**Project administration:** Anton A. Nizhnikov.

**Software:** Kirill S. Antonets, Anna I. Sulatskaya.

**Validation:** Kirill S. Antonets, Mikhail V. Belousov, Anna I. Sulatskaya, Maria E. Belousova, Anastasiia O. Kosolapova, Maksim I. Sulatsky, Elena A. Andreeva, Pavel A. Zykin, Yury V.

Malovichko, Oksana Y. Shtark, Anna N. Lykholay, Irina M. Kuznetsova, Konstantin K. Turoverov, Elena Y. Kochetkova, Alexander G. Bobylev, Konstantin S. Usachev, Igor A. Tikhonovich.

**Visualization:** Kirill S. Antonets, Mikhail V. Belousov, Anna I. Sulatskaya, Anastasiia O. Kosolapova, Maksim I. Sulatsky, Pavel A. Zykin, Alexander G. Bobylev, Konstantin S. Usachev, Anton A. Nizhnikov.

**Writing – original draft:** Kirill S. Antonets, Mikhail V. Belousov, Anna I. Sulatskaya, Maksim I. Sulatsky, Elena A. Andreeva, Pavel A. Zykin, Yury V. Malovichko, Oksana Y. Shtark, Anna N. Lykholay, Oleg. N. Demidov, Anton A. Nizhnikov.

**Writing – review & editing:** Kirill S. Antonets, Mikhail V. Belousov, Anna I. Sulatskaya, Pavel A. Zykin, Alexander G. Bobylev, Anton A. Nizhnikov.

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
