## [Editor Report · Decision Letter 0]

23 Oct 2019

Dear Dr Nizhnikov, 

Thank you for submitting your manuscript entitled "Accumulation of storage proteins in plant seeds is mediated by amyloid formation" for consideration as a Research Article by PLOS Biology.

Your manuscript has now been evaluated by the PLOS Biology editorial staff as well as by an academic editor with relevant expertise and I am writing to let you know that we would like to send your submission out for external peer review.

Please re-submit your manuscript within two working days, i.e. by Oct 25 2019 11:59PM.

Kind regards,

Lauren A Richardson, Ph.D

Senior Editor

PLOS Biology

---

## [Decision Letter · Decision Letter 1]

16 Dec 2019

Dear Dr Nizhnikov,

Thank you very much for submitting your manuscript "Accumulation of storage proteins in plant seeds is mediated by amyloid formation" for consideration as a Research Article at PLOS Biology. Your manuscript has been evaluated by the PLOS Biology editors, an Academic Editor with relevant expertise, and by several independent reviewers.

In light of the reviews (below), we will not be able to accept the current version of the manuscript, but we would welcome re-submission of a much-revised version that takes into account the reviewers' comments. We cannot make any decision about publication until we have seen the revised manuscript and your response to the reviewers' comments. Your revised manuscript is also likely to be sent for further evaluation by the reviewers.

We expect to receive your revised manuscript within 2 months. 

**IMPORTANT - SUBMITTING YOUR REVISION**

*NOTE: In your point by point response to to the reviewers, please provide the full context of each review. Do not selectively quote paragraphs or sentences to reply to. The entire set of reviewer comments should be present in full and each specific point should be responded to individually, point by point.

*Re-submission Checklist*

*Published Peer Review*

*PLOS Data Policy*

*Blot and Gel Data Policy*

Sincerely,

Lauren A Richardson, Ph.D

Senior Editor

PLOS Biology

REVIEWS:

Reviewer #1: 

This manuscript reports that seeds of the garden pea Pisum sativum contain amyloid-like aggregates and that the most abundant storage protein, Vicilin, is able to assemble into amyloid structures. The data support the presence of amyloids in vivo and in vitro. The stability of the amyloid structures changes with seed maturation and assemblies disappear at seed germination.

The authors have identified Vicilin as being resistant to treatment with SDS during isolation from pea seeds. Tryptic digestion followed by mass spectrometry indicates that the full-length protein is present in detergent-insoluble polymers.

Experiments with full-length (FL) or the two isolated cuprin domains indicate that these proteins are able to form fibrils with amyloid character, when incubated in vitro under denaturing conditions. Some fibrillar material is observed under relatively mild conditions, while treatment with HFIP, known to induce secondary structure, leads to the formation if extensive fibrillar structures.

Secretion of the proteins on the surface of the E. curli cells was suggestive of the formation of amyloid-structures aggregates by these proteins. The proteins were also expressed in S. cerevisiae and demonstrates intracellular aggregation. However, many aggregation-prone proteins may form puncta in vivo in yeast. A key reported finding is that vicilin and ThT-positive structures are found co-localised in mature seeds in vivo.

The authors suggest that this is the first demonstration of functional amyloids in plants. This would certainly be a novel and significant finding, since no functional amyloids have been reported in plants to date – although several plant proteins have been shown to form amyloids under denaturing or modifying conditions.

Some work has even suggested that plants may actively produce small molecule inhibitors of fibrillation – although this study suggests that the Vicilin amyloids are associated with storage within seeds and that they disappear upon germination.

However, the work is not described in sufficient detail, nor are the different protein preparations sufficiently well characterised, to fully support the findings as they currently are presented in the manuscript. The experimental design is appropriate and the authors have employed a wide variety of approaches to validate the findings. However, more detail about experimental methods is required to provide confidence in the results.

Points to be addressed:

P2 2nd sentence in 2nd paragraph. Change to” Amyloid deposition is associated with the development… because in many conditions, the direct link between amyloid deposition and pathogenesis is unclear.

Include further mention of many plant proteins that have been shown to be capable of forming amyloid structures under denaturing conditions:

https://www.sciencedirect.com/science/article/abs/pii/S0301462218300139

https://www.sciencedirect.com/science/article/pii/S0014579399007899

The authors should also discuss the early suggestions that plants were rich in amyloids – although it seems likely that these were carbohydrate-rich structures that were characterised in this way, since amyloid-specific stains were not used. See KOOIMAN, P. Amyloids of Plant Seeds. Nature 179, 107–109 (1957) and other associated publications.

No detail is provided regarding the protocol used for refolding proteins from the denaturing conditions used for protein purification. What methods were used to determine that proteins had refolded correctly?

The reference to the Methods in Enzymology protocol is for purification only and does not include not sufficient to verify that proteins were produced correctly and refolded to a native, monomeric state.

What is meant by “Proteins were concentrated by ethanol? Details should be provided. Methanol is used in the MinE reference.

What effect does 50% HFIP have on the CD spectra?

Why does the Cupin-1.2 run so large on the gel in both boiled and unboiled samples? (fig 1c)

Replace “fibrillary” with “fibrillar” throughout. This is a Microsoft Word auto-correct error.

Absorption not Absorbtion in Figure 2d.

While there is some overlap of the signals from ThT and Vicilin (Fig. 5a-c), there are many structures that are strongly ThT-positive but do not show Vicilin co-localisation. What are these structures? Some analysis of the degree of co-localisation should be performed, e.g. with Coloc2 in ImageJ.

The image in Fig 5d-f is difficult to evaluate. A bright field image should be included. The staining of these structures cannot be evaluated at this magnification.

TEM analysis of these isolated, purified protein bodies should be performed and included, since the mild purification should allow retention of fibrillar structures.

SDS PAGE analysis of these purified bodies should be conducted, with silver staining and/or Coomassie to indicate relative content of Vicilin, and other proteins, in these preparations.

Is it surprising that antibodies raised against purified Vicilin react strongly with the amyloid form? They also bind the fragments after proteolytic digestion. It is not clear in the Methods section how the protein was prepared for immunisation. Which protocol was used? How was it verified that the antibodies were specific for Vicilin? If recombinant protein was used, then was it raised against full-length protein?

Further discussion should be included regarding the differences in proteolytic resistance of the material from total protein lysates and fibrils formed from purified recombinant proteins.

What is the likely cause of the protolytic degradation observed from 20 days onwards, i.e. loss of M band?

Further discussion should be included about the likely differences in protease-resistant aggregates between pea seeds and commercial canned products. For example, what processing occurs that could account for the differences?

----------------

Reviewer #2: 

Manuscript by Antonets et al. investigates the amyloid properties of plant proteins. It was previously predicted that the Vicillin protein from the seeds of the garden pea Pisum sativum L shows amyloid like properties. In this study authors go on to show that Vicillin protein that comprises of 2 subunits, Cupin 1.1 and Cupin 1.2 , show amyloid like properties. Authors determine that Cupin 1.2 shows more fibrillar nature while Cupin1.1 and full vicillin protein show aggreagates. Authors conclude using Circular Dichroism, ThioflavinT,microscopy to prove that Vicillin amyloid properties. Although the study is interesting, some of the results and their interpretations do not fully justify the overall conclusions suggested by the authors. More detailed analysis and mechanistic studies showing the residues important for how Vicillin establishes fibrils would strengthen the manuscript. One of the more interesting findings is the cytotoxic nature of the fibrils. These results would benefit from the mechanistic mutations showing abrogation of fibrils lead to abrogation of the cytotoxic nature of Vicillin.

Major comments

1- Once amyloid proteins are polymerized they are resistant to degradation. Even though they are boiled in SDS, they don’t depolymerize into monomers and therefore do not enter a 12% SDS-PAGE gel. The fact that the authors show that the Vicillin enters the gel is counterproductive to their argument that this is an amyloid protein. Can the authors comment on this?

2- Figure 1, the TEM pictures only show fibrillar structures in Cupin 1.2. Have the authors tested other buffers here such as potassium phosphate buffer that supports the formation of amyloid fibrils in vitro?

3- HFIP is used to break the amyloid fibrils into its monomers. The fact that HFIP usage leads to higher ThT levels is puzzling. Are there any other examples in amyloids that show a similar characteristic? It would be good to use a positive control not only for this assay but other assays as well.

4- Figure 2. ThT polymerization curves using recombinant proteins would help the readers to understand the polymerization kinetics of these molecules. Also calculation of lag times could help identify the differences in Vicillin and its subunits. In its current form the ThT experiments are not very informative

5-Manuscript benefit from the comparison of Cupin 1.1 and Cupin 1.2 amino acid sequence comparisons and the mechanistic explanation for why Cupin 1.2 forms fibrils instead of aggregates like vicillin and cupin 1.1. Would the alanine substitution of the predicted residues important for beta sheet fibrillar structure change the amyloid readouts and the structure of these proteins? 

6- Amyloid proteins show around 90% beta sheet conformation. Can the authors comment if Vicillin is showing only 40-50% beta sheets by CD, would it be described as a true amyloid? This can be included in the discussion.

7- As the amyloids adopt a fibrillar structure their toxicity decreases (e.g. beta-amyloid). The fact that the vicillin fibrils show toxicity is biologically interesting. Would disrupting the fibrillar structure mechanistically disrupt the cytotoxic nature?

Minor Comments

1. Abstract would benefit from the description of findings on Cupin 1.1. and 1.2.

2. Supplementary figures 3-6 lack the information whether these images are western blots and which antibodies are used.

----------------

Reviewer #3: 

The manuscript by Antones et al., presents exciting results, showing that similarly to Eukarya bacteria and archea, plant seeds also produce proteins that potentially harbor an amyloid fold. The authors use various commonly acceptable methods to prove this point, including resistance to SDS treatment, ThT and Congo Red binding, cross-seeding, and toxicity for yeast and mammalian cells. Building on their previous bioinformatics work, they have performed an extensive in vitro characterization of the protein Vicilin and they have further showed that ThT binding co-localizes with Vicilin in vivo and that certain forms of Vicilin are toxic for yeast and mammalian cells.

Major comments:

While the authors use a variety of methods to show that plant seeds proteins exhibit an amyloid fold, these methods are rather indirect and not quantitative and they lack some more structural insight that could have been obtained by more quantitative methods such as NMR, and (electron or x-ray) diffraction methods. The data presented in the paper leans heavily on such indirect evidence for amyloid formation, but being the pioneers to come up with the claim that plant seeds produce amyloid proteins, I would expect the (nicely studied) indirect results to be accompanied with stronger structural evidence of amyloid formation.

Additional questions and comments:

1. The CD results (figure 2) are not conclusive. It is very difficult to determine from this data that the proteins exhibit a beta sheet structure, let alone a cross beta sheet structure. Some CD curves (Vicilin) would probably fit better a random coil rather than a beta sheet structure, as is also indicated in the text. To the very least, I suggest to use a few secondary structure analysis software (such as Dichroweb, Biotools) in order to deconvolute the CD spectra and extract the contribution of different secondary structures to each curve. Some percentage of 'ordered beta sheet' is referred to in the text, but it is not clear what is the origin of this number

2. The ThT data is static, but a kinetic study of aggregation is missing as it will allow the reader to compare the aggregation kinetics of the plant proteins with ThT kinetic data obtained with well-studied common amyloid proteins (a beta, alpha synuclein, curli).

3. It is not clear from the text what was the purpose of the fluorescence decay experiments.

4. In the aggregation seeding experiments, it was not clear how seeds were produced and why they were defined as 'seeds'.

5. The Discussion should be more concise and less repeating the results.

---

## [Decision Letter · Decision Letter 2]

18 May 2020

Dear Dr Nizhnikov,

Thank you for submitting your revised Research Article entitled "Accumulation of storage proteins in plant seeds is mediated by amyloid formation" for publication in PLOS Biology. I have now obtained advice from the three original reviewers and have discussed their comments with the Academic Editor. 

Based on the reviews (attached below), we will probably accept this manuscript for publication, assuming that you will modify the manuscript to address the remaining points raised by Reviewers 1 and 3.

We expect to receive your revised manuscript within two weeks. Your revisions should address the specific points made by each reviewer. In addition to the remaining revisions and before we will be able to formally accept your manuscript and consider it "in press", we also need to ensure that your article conforms to our guidelines. A member of our team will be in touch shortly with a set of requests. As we can't proceed until these requirements are met, your swift response will help prevent delays to publication.

*Copyediting*

*Published Peer Review History*

*Early Version*

*Submitting Your Revision*

Sincerely,

Ines

--

Ines Alvarez-Garcia, PhD

Senior Editor

PLOS Biology

Carlyle House, Carlyle Road

Cambridge, CB4 3DN

+44 1223–442810

DATA POLICY:

Many thanks for submitting all the data underlying the graphs in the figures. Please also ensure that figure legends in your manuscript include information on WHERE THE UNDERLYING DATA CAN BE FOUND.

Reviewers’ comments

Rev. 1:

The authors have added additional experimental data and significantly strengthened the evidence for the amyloid nature of these protein aggregates. I support their contention that some functional amyloids are dissociated by hot SDS and also agree that not all amyloids are 90% beta sheet. Many functional amyloids retain large non-beta-folded domains in the fibrillar form, where these domains do not contribute to the cross-beta core structure.

Points to be addressed.

Figure 3. The XRFD of the vicilin fibrils does indeed show the expected cross-beta reflections. However, the samples of Cupin-1.1 and Cupin-1.2 are very salty and the sharp and intense reflections arising from buffer and/or salt in the samples makes it difficult to assign the protein-derived reflections clearly. The authors should collect XRFD data from washed fibrils if possible.

In Figure 5, parts j-m showing TEM images - the scale bar is stated to be 200 microns. Is this correct? Perhaps this should be 200 nm? Otherwise the fibrils are inexplicably large.

Rev. 2:

All the previous critiques were addressed accordingly. The manuscript reads well in its current state and clarity has increased. This is an important study that will increase the understanding about the biogenesis of amyloids and open a new track in discovery of Plant amyloid proteins.

Rev. 3:

The authors have addressed all my concerns/questions and the manuscript has improved immensely. Small comment: Figure 1(e) is lacking a title and units of the x -axis.

---

## [Editor Report · Decision Letter 3]

19 Jun 2020

Dear Dr Nizhnikov,

On behalf of my colleagues and the Academic Editor, Roland Riek, I am pleased to inform you that we will be delighted to publish your Research Article in PLOS Biology. 

Early Version

PRESS 

Kind regards,

Alice Musson

Publishing Editor, 

PLOS Biology

on behalf of

Ines Alvarez-Garcia,

Senior Editor

PLOS Biology